# Spatial and temporal variability of groundwater recharge in a sandstone
# aquifer in a semi-arid region
**Ferdinando Manna [1], Steven Murray [2], Daron Abbey [2], Paul Martin [2,3], John Cherry [1],**
**Beth Parker [1]**
[1] G[360] Institute for Groundwater Research, College of Engineering and Physical Sciences,
University of Guelph, Guelph, Ontario, Canada
[2] Matrix Solutions Inc., Guelph, Ontario, Canada
[3] Aqua Insight Inc., Waterloo, Ontario, Canada.
**Abstract**
With the aim to understand the spatial and temporal variability of groundwater recharge, a high-
resolution, spatially-distributed numerical model (MIKE SHE) representing surface water and
groundwater was used to simulate responses to precipitation in a 2.16 km$^2$ upland catchment on
fractured sandstone near Los Angeles, California.  Exceptionally high temporal and spatial resolution
was used for this catchment modeling: hourly climate data, a 20x20 meter grid in the horizontal plane
and 240 numerical layers distributed vertically within the thick vadose zone and in the upper part of
the groundwater zone.   The finest-practical spatial and temporal resolution were selected to
accommodate the large degree of surface and subsurface variability of catchment features. Physical
property values for the different lithologies were assigned based on previous on-site investigations
whereas the parameters controlling streamflow and evapotranspiration were derived from
calibration to continuous streamflow at the outfall and to average hydraulic heads from 17 wells.
Confidence in the calibrated model was enhanced by validation through, i) comparison of simulated
average recharge to estimates based on the applications of the chloride mass-balance method to data
from the groundwater and vadose zones within and beyond the catchment and, ii) comparison of the
water isotope signature ($^{18}O$ and $^{2}H$) in shallow groundwater to the variability of isotope signatures
for precipitation events over an annual cycle and, iii) comparison of simulated recharge time series
and observed fluctuation of water levels. The average simulated recharge across the catchment for
the period 1995-2014 is 16 mm $y^{-1}$ (4% of the average annual precipitation), which is consistent with
previous estimates obtained by using the chloride mass balance method (4.2% of the average
precipitation).  However, one of the most unexpected results was that local recharge was simulated
to vary from 0 to > 1000 mm $y^{-1}$ due to episodic precipitation and overland runoff effects. This
recharge occurs episodically with the major flux events at the bottom of the evapotranspiration zone,
as simulated by MIKE SHE and confirmed by the isotope signatures, occurring only at the end of the
rainy season. This is the first study that combines MIKE SHE simulations with the analysis of water
isotopes in groundwater and rainfall to determine the timing of recharge in a sedimentary bedrock
aquifer in a semi-arid region.  The study advances the understanding of recharge and unsaturated
flow processes and enhances our ability to predict the effects of surface and subsurface features on
recharge rates. This is crucial in highly heterogeneous contaminated sites because different
contaminant source areas have widely varying recharge and, hence, groundwater fluxes impacting
their mobility.
**Introduction**
Assessment of groundwater recharge is fundamental to create strategies for management of water
resources and to estimate volumetric groundwater flow through contaminated sites. Recharge rates
represent an indication of upper limit of the volume of precipitation that may be accessible for
sustainable use and can govern the volume of water available to transport contaminants. Its
importance is greater in semi-arid regions where dominance of evapotranspiration limits water
resources. In these regions, estimated recharge rates depend on the temporal and spatial resolution
of the investigation and the uncertainties associated with recharge values are usually large
(Scanlon, 2000; Xie et al., 2018; Crosbie et al., 2018). In favorable circumstances, geochemical-based
methods have proven to be especially useful for estimating recharge rates. In areas where the
geologic and anthropogenic sources of chloride in the subsurface are negligible, the distribution of
chloride in the vadose zone and  groundwater has been used to calculate long-term site-wide
(Wood and Sanford, 1995; Gebru and Tesfahunegn, 2018; Jebreen et al., 2018) and location-specific
recharge values (Heilweil et al., 2006; Huang et al., 2018), to determine mechanisms of flow in the
vadose zone (Sukhija et al., 2003; Li et al., 2017), and to evaluate the effects of environmental
changes on recharge (Scanlon et al., 2007; Cartwright et al., 2007). Elevated tritium in precipitation
derived from atmospheric releases during nuclear tests in the 1960's and transported into the
subsurface has also been an invaluable tracer to determine modern recharge and mechanisms of
flow in both vadose and groundwater zones (Cook and Böhlke, 2000; De Vries and Simmers, 2002).
These geochemical and isotopic techniques are based on the interpretation of hydrologic process
influences on the distribution of tracers in the subsurface but cannot show the dynamic, short-term
temporal effects nor provide a continuous spatial representation of these processes at the
catchment scale.
Numerical hydrologic models that integrate surface water and groundwater flows have been
developed to simulate the spatial and temporal distribution of surface runoff, infiltration,
evapotranspiration and groundwater recharge. However, the application of nearly all such
simulation tools have been limited to humid regions (Wheater et al., 2007) with minimal
application to semiarid regions. Scanlon et al.  (2006), in their review on recharge in semiarid areas,
reported only 7 papers providing a continuous spatial distribution of recharge, out of a total of 98
studies. However, these studies investigated large areas, from 1,039,647 km$^2$ (Flint and Flint, 2007)
to 60 km$^2$ (Flint et al., 2001), using a relatively coarse spatial resolution (from 72,900 m$^2$ - Flint and
Flint, 2007 to 900 m$^2$ - Flint et al., 2001).  In the last decade, although modeling techniques have
advanced to include combined surface water-groundwater simulations, recharge in semiarid areas
has been represented with a GIS approach (Hernández-Marín et al., 2018) often using remote
sensing data (Wang et al., 2008; Coelho et al., 2017; Crosbie et al., 2015) or neglecting the surface
water component and focusing on unsaturated zone  (Levy et al., 2017; Turkeltaub et al., 2015).
Among the commercially available models, the physically based MIKE SHE represents the land-
based hydrologic system, with an integration of the surface flows (i.e. precipitation, infiltration,
evapotranspiration and runoff) and subsurface flows (i.e., percolation into the vadose zone and
recharge across the water table) (Ma et al., 2016). However, the literature shows only two
applications of MIKE SHE to assess recharge in semiarid areas. Liu et al. (2007) analyzed the
recharge response associated with overland flow in an alluvial watershed (surface area: 91 km$^2$ -
cell size: 2,500 m$^2$) in the Tarim Basin, China. Smerdon et al. (2009) distinguished and quantified
the contributions of three sources to the total recharge for a valley bottom aquifer in the Okanagan
Basin (Canada) (surface area: 130 km$^2$ - cell size: 10,000 m$^2$).
In this study encompassing a 20-year period (1995-2014), we used MIKE SHE to simulate the
recharge and the other hydrologic processes in a small catchment (2.16 km$^2$) located on an exposed
bedrock upland plateau (from 650 to 490 m asl) in the Simi Hills, near Los Angeles, California (Fig.
1). The area is semi-arid with potential evapotranspiration (CIMIS, 1999) exceeding the average
annual precipitation (396 mm as the recorded average annual precipitation over the 1995 -2014
period). The bedrock consists of sandstone with interbeds of shale and siltstone, densely fractured
with bedding parallel partings and vertical joints and faults (Cilona et al., 2015; Cilona et al., 2016;
Link et al., 1984; MWH, 2016) (Fig. 2). The hydrogeology of the site has been investigated
intensively over the past 20 years because of the chemical contamination (mainly Trichloroethene -
TCE) in groundwater ( Pierce et al., 2018a; Pierce et al., 2018b; Sterling et al., 2005; MWH, 2009;
Cherry J.A., 2009) and construction and application of a 3-D flow model (FeFlow) has been an on-
going effort supporting characterization and corrective measures (AquaResource and MWH, 2007).
From the application at the site of the chloride mass balance (CMB), based on measurement of
chloride in atmospheric deposition, surface water and groundwater, Manna et al. (2016) estimated
a long-term average recharge of 19 mm $y^{-1}$, corresponding to the 4.2 % of the average precipitation
(455 mm for the period 1878-2014). More recently, Manna et al. (2017) analyzed porewater Cl
concentration profiles from the vadose and groundwater zones at 11 locations across the site. This
provided spatially variable, long-term recharge values ranging from 4 to 23 mm $y^{-1}$ and indicated
that, on average, 80% of the flow in the vadose zone occurs as intergranular flow in the rock matrix
and 20% as fracture flow. These chloride-based methods lump together hydrologic processes
providing long-term recharge estimates for only few locations across a large site. However, to
inform the 3-D groundwater flow model and to simulate plume fluxes, information about the spatial
and temporal distribution of recharge is needed.
In this study, we analyze the spatial and temporal variability of recharge in a catchment of the
contaminated site not only to constrain recharge values but also to uncover hydrologic processes
that cause the borehole-scale spatial variability observed in those previous studies (Manna et al.,
2016; Manna et al., 2017). The catchment was chosen because it is representative of the varied
surface and subsurface conditions found throughout the contaminated area and it is believed to be
minimally impacted during the calibration period by the surface water controls measures in place.
Given that the scope of the paper is to simulate the natural conditions, these initiatives are not
considered in our modeling. To better represent the large range of surface and subsurface features
and provide high-resolution representation of the spatial distribution of recharge, we used hourly
climate data, sub-hourly time step  and a fine grid of 400 $m^2$ cells for a total of 5,420 cells. In
addition to the spatial variability, we also examined the seasonal dynamics of the hydrologic
processes by tracking vadose zone water budgets for representative cells of the model. This
analysis helped in understanding the transient conditions that determine the rates of the
hydrologic processes throughout the year. The model was calibrated using measurements of runoff
from instrumented outfall flows and quarterly observations of groundwater levels in 17 wells
distributed across the catchment for the simulated period. Unlike the previous applications of MIKE
SHE in the literature,  the simulation results were also validated through comparison with transient
water levels from shallow wells, comparison with previous independent recharge estimates based
on application of the Chloride Mass Balance (Manna et al., 2016; Manna et al., 2017) and through
the analysis of water isotopes from rainfall and groundwater that indicated the timing of recharge.
Finally, we proposed a conceptual model for various recharge conditions in the fractured sandstone
aquifer based on the results of the MIKE SHE simulation along with findings of previous recharge
studies for the site (Manna et al., 2016; Manna et al., 2017).  In particular, the MIKE SHE simulations
contributed to the conceptual model concerning the role of surface feature variability (e.g.
topography and vegetation) on the hydrological processes whereas the Cl-based studies informed
the flow mechanisms in the underlying portion of the system.

**The site MIKE SHE model**
The MIKE SHE model (Refsgaard, 1995) simulations were conducted at a  sub-hourly time step
using hourly meteorological data measured from 1995 through 2014 on site and from stations
proximal to the study area. A portion of the rainfall is intercepted by the vegetation canopy, from
which evaporation occurs. The remaining water reaches the surface, where it may infiltrate,
evaporate or runoff downslope if depression storage is satisfied. Water infiltrating into the
subsurface may be evapotranspired back to the atmosphere or percolate down to the water table to
become groundwater recharge. Actual evaporation and transpiration were simulated based on the
Kristensen and Jensen Evapotranspiration Model (Kristensen and Jensen, 1975), which considers
potential evapotranspiration estimated using the FAO 56 Penman-Monteith method (Allen et al.,
1998), available soil moisture and the crop characteristics (depth of the evapotranspiration zone,
leaf area index and crop coefficient) in each grid cell (Table 1). When the rainfall exceeds the
infiltration capacity, water is ponded on the ground surface and is available for runoff. The
infiltration capacity in the model is dynamic and a function of the unsaturated hydraulic
conductivity ($K_u$) and the water content properties (i.e., saturation point, field capacity and
permanent wilting point) of the surficial media. To describe the relation between water content,
conductivity and matric potential, the Van Genuchten model is used (Van Genuchten, 1980).The
rate of runoff is simulated using a 2D diffusive wave approximation and is controlled by the
topographic slope, the surface roughness and detention storage. The latter is the volume of water
stored in surface depressions before runoff starts. The unsaturated zone flow is simulated as the
change in soil moisture, resulting from cyclical input (infiltration) and output (recharge and
evapotranspiration). It is modelled as a 1D column using the full Richards equations (Richards,
1931) with  finite difference cells that have variable discretization from the top of the column
(ground surface) to the base of the column (the unsaturated/saturated zone interface). Given the
variable thickness of the vadose zone and the low water fluxes, the model was run several times to
set consistent initial conditions. Our analysis began when the simulation showed that the degree of
change in average recharge value from one run to the next was about 0.3% indicating near steady-
state conditions. Recharge was calculated anytime that infiltration water arrives at the water table.
Most precipitation events do not result in recharge because infiltration into the shallow subsurface
is intercepted and evapotranspired.  The flow in the groundwater zone was represented using 3D
finite difference Darcy equation. A fixed head boundary applied along the lateral sides and the
bottom of the model domain (490 m asl) was used to simulate the flow to and from the deeper
groundwater system, not explicitly represented in the integrated model but which extends several
hundred meters (Fig. 3).  These fixed heads are based on observed groundwater levels at the site
and simulations based on a detailed 3-D groundwater flow model system that includes the
catchment and a much larger domain beyond  (AquaResource and MWH, 2007). The groundwater
contribution to streamflow is minimal and intermittent ($\sim$ 0.1 mm y$^{-1}$ for the period of 1995-2014)
and only occurs at the farthest downstream location of the catchment where the groundwater table
rises close to the ground surface.
*Climate data*
Hourly rainfall data were collected from two stations within the catchment boundaries: the Sage
Ranch station, managed by Ventura County watershed
(http://www.vcwatershed.net/hydrodata/php/getstation.php?siteid=272#top) and the Simi Hills-
Rocketdyne Lab, managed by Boeing Inc. The annual precipitation ranges from 99 mm (2014) to
976 (1998), with an average value of 396 mm $y^{-1}$. The seasonal precipitation regime is
Mediterranean, with 77% of the total precipitation occurring from December to March.
Daily maximum and minimum air temperature observations were obtained from two climate
stations of the NOAA network: from 1995 to 1998 data were gathered from the Cheeseboro station
(https://www.ncdc.noaa.gov/cdo-web/datasets/GHCND/stations/GHCND:USR0000CCHB/detail)
and from 1998 to 2015 from the Van Nuys station (https://www.ncdc.noaa.gov/cdo-
web/datasets/GSOM/stations/GHCND:USW00023130/detail), respectively 6 km SW and 18 km E
of the study site. Temperatures were adjusted using a dry (10 $^0$C $km^{-1}$) and wet (5.5 $^0$C $km^{-1}$)
adiabatic lapse rate based on the elevation change between the SSFL site and the collecting station.
July, August and September are the warmest months with an average daily maximum temperature
of 30.5, 31 and 30.4 $^0$C, respectively whereas February and December are the coldest with an
average daily maximum temperature of 17and 17.4 $^0$C, respectively. Annual average temperature is
16.7$^0$C.
*Surface and subsurface parameters*
The MIKE SHE model was developed employing a 20 by 20 m finite-difference horizontal-plane grid
to represent the surface physical features, a fine vertical discretization of the vadose zone with 240
numerical layers ranging from 0.1 to 1 m thickness and 2 groundwater zone layers, with thickness
variable from 5 to 185 m, to represent vertical variability at, and just below, the position of the
water table (Fig. 3). This resolution was selected as a compromise between representation of
spatial variability at a more detailed scale and reasonable computational time. Maps of topography,
vegetation, surficial geology and land use were used to assign surface parameters (Fig. 1, Fig. 2 and
Fig. 4).  High resolution topographic data (2 feet interval elevation contours) were obtained based
on an aerial survey of the site in 2010. These topography data were used to define the ground
surface elevations (Fig. 1).
The surface and subsurface hydrogeologic units include alluvium, fractured weathered and
unweathered bedrock comprised of sandstone, siltstone and shale beds of varying thickness, grain
size and cementation (Fig. 2 and Fig. 3).  The physical properties of these units, derived from
previous on-site investigations (Allegre et al., 2016; Quinn et al., 2015; Quinn et al., 2016) and
adjusted by calibration, are summarized in Table 2.  In particular, our model uses three separate
sets of Van Genuchten parameters to represent the pressure saturation-hydraulic conductivity
relationships. The parameters used reflect our understanding that the rock matrix transmits the
largest volume of recharge (80%), while recharge through the fractures is minimal (20%) (Manna
et al., 2017). Therefore, the relationships used are biased towards the matrix response. These
values were further calibrated using the groundwater level responses and the streamflow. Further
rock core samples indicate a high moisture content (~80%) (Cherry et al., 2009) indicating that $K_u$
is often close to $K_s$ and the hydraulic conductivity-saturation curve reflects this understanding.
Four land use classes were identified and delineated based on aerial imagery and local land cover
datasets (Davis et al., 1998): developed areas (roads, building, parking lots); chaparral (chamise,
scrub oak), coastal scrub (Black sage) and exposed bedrock (areas without vegetation) (Fig. 4). The
first category represents only 5% of the study catchment whereas the two vegetation classes
(chaparral and coastal sage scrub) cover 83% of the area. The remaining 12% is represented by
bedrock outcrop. This latter category was subdivided into two classes: non-massive bedrock and
massive bedrock based on physical appearance. Massive bedrock areas were identified based on
rock masses that have resisted erosion over the decades and are presumed to be poorly-fractured
and/or well cemented such that local infiltration through these rock units is very low.  These cell
assignments were identified using topography and imagery analysis. First, we used the minimum
downslope elevation change approach to identify topographic ridges; this algorithm calculates the
minimum elevation drop to a downslope neighbor. In a second stage, we isolate from the land use
map the exposed bedrock areas. Vegetation, indeed, generally does not grow on well cemented
rock. Finally, massive bedrock areas were assigned cells with downslope elevation change greater
than 1.25 meters in areas without vegetation.
Values of Leaf Area Index, depth of the root zone, surface roughness and Manning's number were
assigned to each land use class-specific parameters, based on the calibration process, with final
values similar to those available in the literature (Canadell et al., 1996; Scurlock et al., 2001; Chin et
al., 2000) (Table 1).  To calculate the actual evapotranspiration, a crop coefficient varying monthly
between 0.53 and 1.02 has been used. This estimate are based on i) reference crop
evapotranspiration rates (RET) for Zone 9 of the Reference Evapotranspiration Zones map of the
California Irrigation Management Information System, that corresponds with the area the site is
within (ITRC, 2003), ii) a 'Pasture and Misc. grasses' land class chosen as representative of the site
and iii) a reduction of 8% to account for bare spots in vegetation and reduced vigor (ITRC, 2003).
*Unsaturated zone water budgets*
To assess the temporal variability of infiltration, evapotranspiration, change in storage and
recharge, we extracted the simulated unsaturated zone water budgets for two locations
representing the span of variability of the catchment. The two locations were selected based on
surface geology (Fig. 2) and land use category (Fig. 4):  UZ1 represents an area of outcropping
bedrock without vegetation and UZ2 represents a cell with alluvium and vegetation cover. The
average infiltration value over the simulated period at the two locations (UZ-1: 87 mm y$^{-1}$; UZ-2:
395 mm y$^{-1}$) matches the average infiltration value for all the cells of the catchments with same land
use and surface geology. For these cells, we extracted the weekly time series of infiltration,
evapotranspiration, storage variations and flux at the bottom of the ET zone (i.e., drainage). The
latter indicates the volume of water that infiltrates into the vadose zone and will eventually become
recharge upon reaching the water table. The analysis of the seasonal variability of these fluxes
provided insights about their transient nature and about the effect of the surface variability on the
hydrologic processes in the unsaturated zone.
*Approach for model calibration*
In the model calibration procedure, the simulation results were compared to observed processes
and, to obtain acceptable matches, 10 parameters were available to adjust: surface roughness,
detention storage, imperviousness, rooting depth, leaf area index (LAI), crop coefficient,
unsaturated hydraulic conductivity and water content parameters of alluvium and weathered
bedrock. These were tested against an objective function of streamflow and groundwater level
measurements. An objective function is a measure of overall model fit of simulated to observed
values of groundwater levels and streamflow.
For the streamflow calibration, we compared the surface runoff generated by MIKE SHE to the data
collected at the catchment outfall between 2009 to 2011. This time interval had minimal
occurrence of substantial anthropogenic activities and was representative of natural hydrologic
conditions, as reported also by Manna et al. (2016).
For the calibration to groundwater levels, quarterly manually measured water level data were used.
Excluded from the calibration data were: i) wells with screened interval below the bottom of the
model domain (490 m a.s.l.), and ii) wells where the water table is strongly influenced by
subsurface complexity not represented in the saturated zone portion of the MIKE SHE model. After
these exclusions, water level data from 17 wells being used with water depths ranging from 25 to
137 meters bgs (Fig. 1, 2 and 4).  The number of measurements in the time series at each well
varies from 1 (RD-130) to 139 (WS-09B) measurements. In the calibration procedure, average
values were used for comparison with average simulated values.
The calibration process proceeded in an iterative manner. After each calibration run, the two
calibration targets were examined with a variety of metrics. For the streamflow, we analyzed mean
error for simulated and observed average annual flow; mean error, root mean squared error,
correlation and Nash Sutcliffe Efficiency for the simulated and observed average monthly and daily
flows. An additional qualitative measure of the correlation between precipitation and streamflow
event was provided by the analysis of the graphical of plots of observed and simulated daily
streamflow hydrographs.
For the groundwater levels, the metrics were mean error, mean absolute error, root mean squared
error and normalized root mean squared error for the simulated and observed average water
levels. In addition, residual plots of simulated and observed water levels provided a quantitative
and qualitative assessment of the residual error present at the observation well throughout the
domain. Spatial patterns of groundwater level residual were compared against other spatial data
(e.g. hydraulic conductivity, boundary conditions, land uses, surface geology) to evaluate potential
correlations and adjustments that could improve the calibration.
Following an assessment of these calibration targets, the ten model parameters were adjusted for
better calibration metrics. In instances where the results were not consistent with the site
conceptualization, consideration was given as to whether an alternative conceptualization would
explain the results predicted by the model. Testing of alternative conceptualizations through
manual simulations was chosen over the alternative method of optimization of a single
conceptualization using software such as PEST (Doherty, 2004) given the uncertainty in how to
parameterize models in these semi-arid environments. Given the structural changes
(representation of the unsaturated flow, representation of impervious areas) that were made to the
model during the several simulations, it was not possible to carry out an exhaustive optimization or
sensitivity analysis. However, through the calibration process we gained semi-quantitative
information about the model sensitivity to each parameter which is presented in the results section.

*Approach for model validation*
To obtain confidence about the reasonableness of the results, simulation results from the calibrated
model were tested by a validation procedure, which included comparison to previous independent
recharge estimates based on chloride, and timing of recharge from isotopic data ($^{18}O$ – $^{2}H$) and from
analysis of observed fluctuations of water level hydrographs, not used in the calibration. The
premise of the validation is that the calibrated model must provide results consistent with the
validation information, that are entirely independent of the parameter assignments made in the
calibration.
Manna et al. (2016) estimated an average long-term recharge of 19 mm y$^{-1}$ for the same catchment
using the chloride mass balance (CMB) method, based on the average Cl concentration measured in
the atmospheric deposition, comprised of rainfall and dry fallout (2.6 mg L$^{-1}$), surface water at the
catchment outfall (4 mg L$^{-1}$) and groundwater (52.5 mg L$^{-1}$).  Since chloride concentration in
groundwater is proportional to the concentrating effect of water loss due to evapotranspiration, it
can be used as a proxy to determine the range of variability in recharge. Chloride concentration in
shallow groundwater monitoring wells ranges across the area from 17 to 162 mg/L corresponding
to recharge values of 43 and 5 mm y$^{-1}$, respectively.  Manna et al (2017) also provided insights
regarding spatial variability of recharge within the catchment based on analysis of Cl profiles in
porewater from the vadose zone and groundwater which indicate a range of recharge from 4 to 21
mm y$^{-1}$ corresponding to <1 – 4.7% of the average annual precipitation for 4 locations located
within the catchment area. Although the recharge values obtained from the CMB method integrate
hydrologic processes occurring over longer time, from decades to millennia, they represent a
reasonable assessment of long-term, site-wide and location-specific average values and are
valuable for validation purposes.
Samples of rainfall and groundwater were analyzed for water isotopes ($^{18}$O – $^{2}$H) . These water
isotopes are commonly used to assess evaporative processes and to determine sources and origins
of different groundwaters. Typically, the water isotope values vary seasonally over the annual cycle,
so that the groundwater composition reflects the season with most of the recharge. In this study,
we compared the isotopic signature of groundwater to that of precipitation for an entire
hydrological year to determine whether the timing of recharge indicated by the model is consistent
with the isotopic signature for the same period of the year. The available isotope data for rainfall
were determined for the period October 1994 to June 1995 collected at two rain gauge stations
(B/886 and RMDF), 5 km from the studied watershed and analyzed in the same year by an
automated gas-source mass spectrometer at the University of California Berkeley. The groundwater
samples were collected from monitoring wells in the studied catchment in two rounds of sampling:
the first in 2003-2004 and the second in 2013 (Fig. 1).
Furthermore, to test the ability of the model to simulate unsaturated zone flow processes and to
reproduce the transient recharge conditions, we compared the simulated time series of recharge,
obtained from MIKE SHE, with quarterly water level measurements at five locations not used in the
calibration process. The depth to groundwater at these wells ranges between 2 and 60 m with
seasonal fluctuations due to the recharge events.  The recharge time series is obtained, extracting
the average, catchment-wide, monthly recharge values.

**Simulation results**
*Model calibration and sensitivity*
Streamflow measured at the outfall occurs in response to rainfall; however, some precipitation
events are followed by very low or no measurable flow (Fig. 5). This is evident for precipitation
events from April to June 2009, October and November 2010 and May and June 2011. In all these
cases, the surface runoff, generated by the precipitation events, infiltrates into the subsurface
without reaching the surface outfall (Fig. 5). These hydrologic dynamics are well simulated by MIKE
SHE. The comparison between the observed and the simulated hydrographs shows a good
correlation for the calibration period ($R^2$=0.97; average difference 4.7%). The average simulated
flow is 48 mm $y^{-1}$, about 14.5% of the average precipitation for the 2009-2011 period (331 mm)
and is almost coincident with the measured flow (46.2 mm $y^{-1}$) (Fig. 5). This value reflects the
precipitation conditions of the 2009-2011 period and is lower than the average runoff over the
entire simulated interval (110 mm $y^{-1}$, 28% of the annual precipitation). Monthly and daily Nash
Sutcliffe Efficiency (NSE) values of 0.94 and 0.87 were achieved respectively, indicating good fit to
observed flows (NSE=1 corresponds to a perfect match).
In addition to the surface water leaving the catchment, the model was also calibrated to the
observed average groundwater head data (Fig. 6). A good match was obtained for the 17 locations,
with almost all values falling within the 10 m confidence interval bands, with a correlation
coefficient of 0.96 and a mean absolute error of 4.5 m (Fig. 6). This good correlation provides
confidence about the spatial distribution of model parameters.
Of the 10 adjusted parameters, unsaturated hydraulic conductivity and water content parameters
of alluvium and weathered bedrock had the strongest effect on the calibration and are, therefore,
well constrained by the measured streamflow and groundwater levels. These geologic features
represent the upper layers of the model domain and variations in their physical and hydraulic
properties control the rate of infiltration, evapotranspiration, drainage and, therefore, recharge. A
third parameter important in the calibration was the detention storage. This is because a
substantial amount of water from precipitation, especially at the beginning of the rainy season,
infiltrates without generating runoff events at the outfall (Fig. 5). This volume of water is controlled
not only by the properties of unsaturated zone (Table 2) but also by the value of detention storage
assigned to each land use class (Table 1). Conversely, alterations in rooting depth, LAI and crop
coefficient only resulted in small changes in streamflow.  This is because significant runoff events
tend to occur during brief high-intensity precipitation events with a magnitude that far exceeds the
relative amount of evapotranspiration, which might occur during these events. For the same reason,
though, these factors had a relatively greater effect on the volume of water available for drainage
and subsequent recharge.

*Spatial variability*
To study the spatial variability of the water budget components, average annual maps of infiltration
(Fig. 7a), evapotranspiration (Fig. 7b) and recharge (Fig. 7c) for the period 1995 – 2014 were
created.
Average infiltration for the catchment is 254 mm $y^{-1}$, corresponding to 64% of the total
precipitation but single cell values span over three orders of magnitude from 9 to > 1000 mm $y^{-1}$
(Fig. 7a). Low infiltration values are found in developed/paved (average 51 mm $y^{-1}$) and massive-
bedrock (average 14 mm $y^{-1}$) cells. Due to the low infiltration capacity, more runoff is generated in
these cells and, thus, infiltration is higher in nearby cells that receive the surface water.  Where
these neighboring cells are covered by alluvium at the surface, infiltration is even higher. On
average, cells with alluvium at the surface have an infiltration value of 332 mm $y^{-1}$, 25% more than
those where bedrock outcrops.  Higher infiltration is also displayed in depressed areas such as
those along the main drainages and where closed topographic depressions occur. These cells collect
most of the surface runoff creating conditions for focused infiltration and recharge.
Only a small portion of water that enters the subsurface reaches the water table because the
majority is lost due to evapotranspiration (Fig. 7b). The average evapotranspiration estimated
using MIKE SHE is 265 mm $y^{-1}$, a value slightly higher than the average infiltration.  This excess of
ET over infiltration is attributed to canopy interception and evaporation of temporarily ponded
surface water. When removing these two water-loss processes, the average evapotranspiration is
237 mm $y^{-1}$, which corresponds to 60% of the annual precipitation and to 94% of the total
infiltration.  Transpiration is the main process of ET contributing to about 70% of the total ET. This
result is expected considering the considerable depth of the roots (up to 5 meters for Chaparral)
and the fact that vegetation covers 83% of the catchment area. Single cell values of ET span over
three orders of magnitude, from 50 to >1000 mm $y^{-1}$.  Since the actual evapotranspiration depends
strongly on the availability of subsurface water, the spatial variability mimics the infiltration
pattern and the two factors are strongly correlated ($R^2$=0.84). Therefore, low ET is associated with
developed (asphalt, buildings) and massive bedrock areas and high ET values are found along the
main surface drainages where infiltration is high and locally available for evapotranspiration. The
presence of alluvium at the surface increases the ET values on average by 25%; for example,
average ET in cells with chaparral and alluvium is 400 mm $y^{-1}$ whereas where chaparral is rooted in
weathered bedrock is ~300 mm $y^{-1}$.
A map of the spatial distribution of the average annual recharge is shown in Fig. 7c. The average
recharge value for the catchment is 16 mm $y^{-1}$ equal to 4.1 % of the precipitation and 6.5 % of the
infiltration.  The range of variability of recharge is over three orders of magnitude and spatially
variable depending on topography, surface geology and land use.  It is noteworthy that 79% of the
catchment has recharge less than 10 mm $y^{-1}$ and 90% less than 30 mm $y^{-1}$, which indicates that the
largest volumes of recharge are focused in small portions of the site. The recharge map (Fig. 7c)
shows the influence of the surface parameters on recharge estimates. Recharge is high along the
main drainage because of the contribution of surface water flowing from the surrounding slopes
and enhanced infiltration where the topographic slope decreases abruptly. Relatively higher
recharge values are also observed in areas with alluvium at the surface because the infiltration and
retention capacities are higher and, therefore, water can seep from the overburden into the bedrock
once the evapotranspiration demand and driving forces are met. Recharge is also higher in cells
without vegetation cover, compared to other cells with equivalent topographic slope and surficial
geology, because the evapotranspiration in these areas is lower.

*Temporal variability*
The seasonal variability of the hydrologic processes was examined analyzing unsaturated water
budgets at two locations with different land use and surficial geology (UZ-1 and UZ-2 in Fig.1)
Among the 20 years, we show the monthly average daily values from 2005 to 2007.  This time span
features a wet year (2005 – 978 mm), a dry year (2007 – 149 mm) and one year with average
precipitation (2006 - 331 mm) and therefore is reasonably representative of the simulated period.
For areas with bedrock outcrop not covered by vegetation (UZ-1 in Fig. 1), the infiltration ranges
from 0 to 2.5 mm d$^{-1}$ (Fig. 8). The infiltration pattern shows null or minimal values during the
summer and positive events during the wet season. Water that enters the subsurface between April
and January replenishes the water content in the ET zone and becomes available for evaporation
but not for drainage.  Evaporation is null during the summer because of the lack of precipitation
and because all the water stored in the first 20 cm of bedrock has been taken up by evaporation in
the previous months. Downward flux at the bottom of the ET zone (i.e. drainage) only happens
episodically when the water content in the ET zone is above the field capacity, at the end of the wet
season (i.e., March and April) or occasionally after exceptionally high-intensity precipitation events
(i.e., January 2005).
For areas with alluvium at surface (UZ-2 in Fig. 1) the infiltration has the same pattern but a
different order of magnitude (from 0 to 30 mm d$^{-1}$) due to the higher infiltration capacity of the
alluvium (Fig. 8). Here, the available water capacity of the ET zone is greater because of the
different physical properties (e.g. larger porosity) of the soil and the greater depth of the ET zone.
Therefore, almost all the infiltration water is taken up by the evapotranspiration. Unlike areas
without vegetation, evapotranspiration is not directly related to precipitation events and occurs
more continuously throughout the year. This is because alluvium stores a greater volume of water
in the ET zone that is nearly completely consumed by ET. A drainage flux is observed only during
high-intensity precipitation events that create near-saturation conditions such that water cannot be
held by tension in the shallow unsaturated zone and downward flow is initiated.
For both cases, drainage is not steady throughout the year but occurs episodically, controlled by
antecedent soil water content in the ET zone and by the intensity of precipitation.  During drier-
than-average years, such as 2007, drainage occurs in areas without vegetation, whereas no
drainage is observed in cells with vegetation cover.  After crossing the bottom of the ET zone, water
arrives at the water table with a time lag depending on the magnitude of the flux and on the
physical properties and the thickness of the vadose zone.
*Model validation*
The ability of the model to simulate transient hydrologic conditions was investigated through the
comparison between well hydrographs at five locations and the temporal variability of recharge
(Fig. 9). The spatially-average recharge rates obtained from MIKE SHE (monthly time-step) range
from 0.95 mm (November 2014) to 9.1 mm (March 2005). The latter is the response to the
extraordinary rainy season that occurred between December 2004 and March 2005 (903 mm)
whereas the first is due to dry conditions of the recent drought in California. The range of depth to
groundwater from 1995 to 2014 at the five locations considered is 2.8 – 14.4 m at RD-09, 17.8 – 30
m at RD-35A, 16.2 – 28.7 at RD-73, 37.7 – 50.8 m at RD-36B and 33.1 – 60.1 at WS-09B. The shape
of these hydrographs depends on surface (surface geology, topographic slope, land use) and
subsurface (mechanisms of flow in the vadose zone) conditions.  For our validation purpose, it is
noteworthy that, at all the locations, the hydrographs show a good match with the recharge time
series such that the peaks in recharge coincide with water table rises. The greatest rises overlap the
two highest recharge periods (1998 and 2005), whereas a constant declining trend is observed
from 2011 to 2014 in response to drier conditions (Fig. 9). The good correlation suggests that, at
this scale, the equivalent porous media approach used is reasonable to simulate average responses
in groundwater because, although the bedrock has many interconnected fractures, it is only a minor
contributor to recharge.
The average recharge value for the catchment from the simulation is 16 mm $y^{-1}$ and is consistent
with previous recharge estimates obtained for the site using the CMB method (19 mm $y^{-1}$ – 4.2% of
the average precipitation, Manna et al., 2016; 16 mm $y^{-1}$ – 3.5% of the average precipitation, Manna
et al., 2017). The frequency distribution of recharge values from the MIKE SHE simulation (92% of
the domain has average recharge lower than 40 mm $y^{-1}$) also corresponds well to the range of
variability based on chloride (from 0 to 43 mm $y^{-1}$) reported by Manna et al. (2016) and Manna et
al. (2017).
For additional information on recharge processes, we analyzed water isotopes obtained from
rainfall and groundwater samples (Fig. 10). The samples show a substantial isotopic range from
one precipitation event to another over the one-year collection period.  $^{18}O$ varies between -2.8 and
-12.1‰ for B/886 and -2.8 and -11.7‰ for RDMF and $^{2}H$ varying between -11 and -89‰ for B/886
and -12 and -85‰ for RDMF (Table 3).  This large range of values is probably due to the two
different trajectories of the precipitation events in southern California, one originating in the Pacific
and one over the Gulf of Mexico, as found by Friedman et al. (1992). The volume weighted mean
values for the two stations are -8.2 and -54.2‰ for B/886 and -8.2 and -56.2‰ for RDMF and are
consistent with global-scale maps of water isotopes for precipitation in southern California (Bowen
and Revenaugh, 2003).
Unlike rainfall, groundwater samples fall within a narrower range: from -6.5 to -7.5‰ for $^{18}O$ and
from -40.2 and to -52.2‰ for $^{2}H$. All the samples are aligned along the local meteoric water line
(Fig. 10) indicating little if any evaporation from standing water on surface. This lack of
concentration effect on the isotopes is apparently in contrast to the chloride data. Manna et al.
(2016) found that Cl concentrations in groundwater are, on average, 20 times greater those from
atmospheric deposition because of the strong influence of evapotranspiration. The common
explanation for the lack of evaporation effects on the water isotopes in groundwater is  that the
transpiration is the main evapotranspiration process ( Clark, 2015;Cook and Böhlke, 2000).
Although transpiration through the vegetation causes a concentration effect on Cl, it does not cause
fractionation of the water isotopes and therefore the groundwater samples are not enriched (Clark,
2015; Cook and Böhlke, 2000).
The lack of evaporative water isotope signature associated with high groundwater Cl concentration
can also be explained by recharging water that crosses the ET zone mobilizing precipitated salts but
without any evaporation. This hypothesis supports the results of the MIKE SHE simulations, which
show that throughout the year there are only episodic fluxes at the bottom of the ET zone (Fig. 9).  A
relevant observation that corroborates this hypothesis is that the isotopic composition of
groundwater is similar to that found in rainfall samples collected at the end of the wet season
(March and June) or, on occasion, with high-intensity precipitation events (January - 203 mm)
(Table 3). This similarity can be attributed to the preponderance of recharge occurring at these
times and thereby resulting in the groundwater values being different from the weighted mean
precipitation by 1.2‰ $^{18}$O and 3‰ $^2$H.
This proposed model of episodic flow through the unsaturated ET zone is also corroborated by the
evidence presented by Manna et al., (2017) that, on average, 20% of the flow in the vadose zone
occurs as fast flow through the interconnected fractured network.

**Discussion and conceptual model for recharge**
To summarize the findings of this study, and its relationship to the literature and to the previous
recharge studies at the site (Manna et al., 2016; Manna et al., 2017), we propose the following
process-based conceptual model for site recharge (Fig. 11).
The average recharge value is 16 mm y$^{-1}$ which is consistent with previous estimates at the site, and
with those obtained for other sandstone aquifers in semi-arid areas in the United States (4% -
Heilweil et al., 2006) and other studies in semi-arid regions around the world (0.2 – 35 mm y$^{-1}$ equal
to 0 – 5% of the average precipitation, Scanlon et al., 2006). Recharge varies greatly across the
catchment as a function of topography, surface geology, and land use. High recharge occurs where
most runoff water seeps into the subsurface, creating conditions for focused recharge. This
condition happens where closed depressions occur and where sloped topography abruptly
transitions to flat along the main surface drainages (Fig. 11a).  In most areas, alluvium covers the
fractured porous bedrock, thus enhancing infiltration and temporary storage of infiltrated water.
Generally, in semiarid regions, high recharge values along a valley, at the edge of the slope referred
to as Mountain Front Recharge (MFR) (Wilson and Guan, 2004). However, our catchment is located
on the top of a ridge standing 300 m above the surrounding valleys (Manna et al., 2016) and, thus,
our case study represents groundwater recharge on the mountain block rather than MFR.
Nonetheless, it is interesting that the processes observed in our small catchment are similar to
those described for aquifer-scale recharge studies (Aishlin and McNamara, 2011; Carling et al.,
2012; Manning and Solomon, 2003; Bresciani et al., 2018) and defined as MFR.
Infiltration from April to December (dry season) contributes to replenish the water content in the
ET zone and remains available for evapotranspiration (Fig. 11b). Conversely, during the wet season,
infiltration crosses the bottom of the ET zone (i.e. drainage) and migrates deeper through the
vadose zone. This happens when the soil is above the field capacity (FC), which is more frequent at
the end of the wet season in March or April and/or during high-intensity precipitation events, (Fig.
11c). This recharging water quickly crosses the ET zone, as shown by the ET zone water budgets
extracted from MIKE SHE (Fig. 9), and by the lack of evaporative signature in isotope composition
(Fig. 10).
The occurrence of this fast/preferential flow out of the ET zone is also corroborated by the analysis
of vertical chloride porewater concentration profiles in the unsaturated zone (Manna et al., 2017).
The Cl concentration is high in the ET zone (up to 10,000 mg L$^{-1}$) and considerably lower in deeper
vadose and groundwater zones (average 49 mg L$^{-1}$). The higher Cl concentrations in the shallow
subsurface is the effect of strong evapotranspiration that takes up water but not chloride, whereas
the lower concentration below is due to fast/preferential flow of water that escapes the
concentrating effect of water loss in the shallower zone. Case studies showing similar results for
water that crosses the ET zone preferentially in time and space to become potentially recharge have
been also reported in literature (Kurtzman et al., 2016), also referred to as selective recharge (Gat
and Tzur, 1967; Florea, 2013; Krabbenhoft et al., 1990) . The occurrence of these fluxes has been
also analyzed in function of precipitation characteristics and antecedent water content with rainfall
intensity being the main factor (Allocca et al., 2015; Crosbie et al., 2012; Nasta et al., 2018; Taylor et
al., 2013).
Upon reaching the deeper vadose zone, water is redistributed between intergranular matrix flow
and fracture flow due to wettability and saturation concepts. The fractures and the matrix pores
drain the water from the ET zone. Active flow through the fractures is possible under conditions
such as ponding or intense precipitation, when a continuous slug of water lets i) the advective front
move ahead into the fracture (1 in Fig. 11c); ii) the matrix water flow into the fractures (2 in Fig.
11c). Otherwise, water is drawn from the fractures into the unsaturated matrix blocks (3 in Fig.
11c) and contributes to the slow vertical intergranular matrix flow (4 in Fig. 11c). According to
Manna et al. (2017), the first two mechanisms are much less frequent and contribute, on average, to
only 20% of the total recharge. It is most likely that conditions for flow in the fractures occur
episodically in areas of the site with high infiltration (topographic low and alluvium at the surface)
where temporary perched systems are observed.

**Conclusions**
For the upland bedrock catchment, the surface water-groundwater numerical model (MIKE SHE),
using a fine numerical grid (20 ×20 m) with calibration to streamflow and groundwater levels,
simulated the spatial and temporal variability of recharge across a 2.16 km² catchment in southern
California, USA.  This is the first study that combined MIKE SHE simulations supported by analysis
of water isotopes and chloride mass balance to assess recharge in a sedimentary bedrock aquifer in
a semi-arid region. The calibrated simulations, indeed, were judged to be reliable and strongly
reflective of the natural system, based on the validation comparisons to mean recharge obtained
independently from the chloride mass balance method (Manna et al., 2016; Manna et al., 2017) and
to the timing of major recharge events indicated by water isotopes and water level fluctuations. The
simulations showed that major flux events at the bottom of the evapotranspiration zone, that result
in recharge tens of meters below the surface, occur episodically mostly at the end of the rainy
season and that recharge varies across the catchment between 0 and 1000 mm y$^{-1}$.  The fine
numerical grid in the horizontal plane allowed meaningful examination of recharge spatial
variability. A substantially coarser grid would obscure influences of key surface features on the
hydrologic processes.
The results obtained from the catchment-scale simulations are being used to specify rules for
recharge to be assigned to the upper boundary condition of a 3-D numerical EPM groundwater flow
model (FeFlow), covering the studied catchment and a much large area beyond (52 km$^2$). The
modeled groundwater domain has many contaminant plumes and recharge is key to determine the
fluxes available to transport contaminants.
The aim of the MIKE SHE model is to represent the natural hydrologic conditions, after site
industrial operations ceased more than a decade ago. During historical operations from 1950's
through mid-2000's, use of imported and pumped groundwater likely caused increases to
infiltration and recharge locally in some areas.  These conditions are beyond the scope of this paper
but worth further consideration in a follow-on study as it relates to when contaminant releases
occurred and may provide insights regarding how contaminant migration rates may have been
influenced. Future modeling efforts will also evaluate the effect on recharge of the surface water
control systems currently in place on the site. These storm water management measures aim to
limit the volume of water leaving the catchment and, therefore, will likely influence the natural
rates of the other hydrologic processes.

**Acknowledgements**
Funding for this work was provided by an NSERC Industrial Research Chair (n. IRCPJ 363783) to
Professor Beth Parker in partnership with the Boeing Company. Field work was supported by
the site owner, their consultants (MWH Inc., now Stantec), and University of Guelph colleagues,
especially Amanda Pierce from the G[360] Institute for Groundwater Research, who collected and
analyzed isotope samples.

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

*Table 1 Land use class-specific parameters to model runoff and evapotranspiration.*


| Land Use Class | Surface roughness (Manning's n) | Detention storage (mm) | Leaf Area Index | Depth of the evapotranspiration zone (m) |
|---|---|---|---|---|
| Developed* | 0.04 | 1 | - | 0.2 |
| Coastal Scrub | 0.2 | 7.5 | 1.8 - 3 | 1.8 - 3 |
| Chaparral | 0.2 | 7.5 | 2.8 - 4.5 | 3.1 - 5 |
| Exposed Bedrock/ Massive bedrock* | 0.05 | 3 | - | 0.2 |




*Table 2 Physical properties of the different hydrogeologic units.*

| Hydrogeologic unit | Lithology | $K_s$ (m s$^{-1}$) | Saturation ($\theta_s$) | Field capacity ($\theta_{fc}$) | Residual Water content ($\theta_r$) | Van Genuchten parameters | | |
|---|---|---|---|---|---|---|---|---|
| | | | | | | $\alpha$ | n | l |
| Alluvium | | $1\times10^{-6}$ | 0.4 | 0.25 | 0.05 | 0.021 | 1.61 | 0.5 |
| Weathered bedrock | | $2\times10^{-7}$ | 0.2 | 0.11 | 0.01 | 0.033 | 1.49 | 0.5 |
| Unweathered bedrock | Shale/Siltstone | $4.1\times10^{-10}$ to $2.3\times10^{-7}$ | 0.13 | 0.1 | 0.025 | 0.01 | 1.23 | 0.5 |
| Unweathered bedrock | Sandstone | $1\times10^{-10}$ to $1\times10^{-5}$ | 0.13 | 0.09 | 0.01 | 0.01 | 2 | 0.5 |

| Unweathered bedrock | Fault zone | 1×10⁻⁹ to 1×10⁻⁶ | 0.13 | 0.1 | 0.025 | 0.01 | 2 | 0.5 |
|---|---|---|---|---|---|---|---|---|




*Table 3 Stable isotope composition of rainfall.*

| Date | B/886 Rain Gauge | | | RMDF Rain Gauge | | | Average | | |
|---|---|---|---|---|---|---|---|---|---|
| | $\delta^{18}O$ | $\delta^2H$ | Rainfall (mm) | $\delta^{18}O$ | $\delta^2H$ | Rainfall (mm) | $\delta^{18}O$ | $\delta^2H$ | Rainfall (mm) |
| 4/10/1994 | -4 | -19 | 3 | | | | -4.0 | -19.0 | 3 |
| 25/11/1994 | -5.2 | -18 | 6 | -5.1 | -16 | 6 | -5.2 | -17.0 | 6 |
| 13/12/1994 | -5.4 | -23 | 9 | -5.4 | -25 | 9 | -5.4 | -24.0 | 9 |
| 24/12/1994 | -10.3 | -77 | 18 | -10.1 | -69 | 18 | -10.2 | -73.0 | 18 |
| 4/1/1995 | -10.3 | -75 | 94 | -9.9 | -69 | 121 | -10.1 | -72.0 | 108 |
| 11/1/1995 | -6 | -33 | 205 | -7.4 | -45 | 202 | -6.7 | -39.0 | 203 |
| 13/01/1995 | -4.4 | -19 | 20 | -4.2 | -20 | 18 | -4.3 | -19.5 | 19 |
| 16/01/1995 | -2.8 | -11 | 12 | -2.8 | -12 | 10 | -2.8 | -11.5 | 11 |
| 26/01/1995 | -12.1 | -89 | 152 | -11.7 | -85 | 150 | -11.9 | -87.2 | 151 |
| 7/3/1995 | -6.8 | -43 | 119 | -6.4 | -40 | 109 | -6.6 | -41.5 | 114 |
| 13/3/1995 | -7.5 | -44 | NA | -7.8 | -45 | NA | -7.7 | -44.5 | NA |
| 24/3/1995 | -5.8 | -22 | NA | -5.5 | -19 | NA | -5.7 | -20.5 | NA |
| 18/5/1995 | | | | -6.4 | -42 | 34 | -6.4 | -42.0 | 34 |
| 22/6/1995 | -8.6 | -62 | 14 | -8.6 | -57 | 14 | -8.6 | -59.5 | 14 |
| Volume weighted mean and total rainfall | -8.2 | -54.2 | 650 | -8.2 | -56.2 | 691 | -8.3 | -55.2 | 689 |


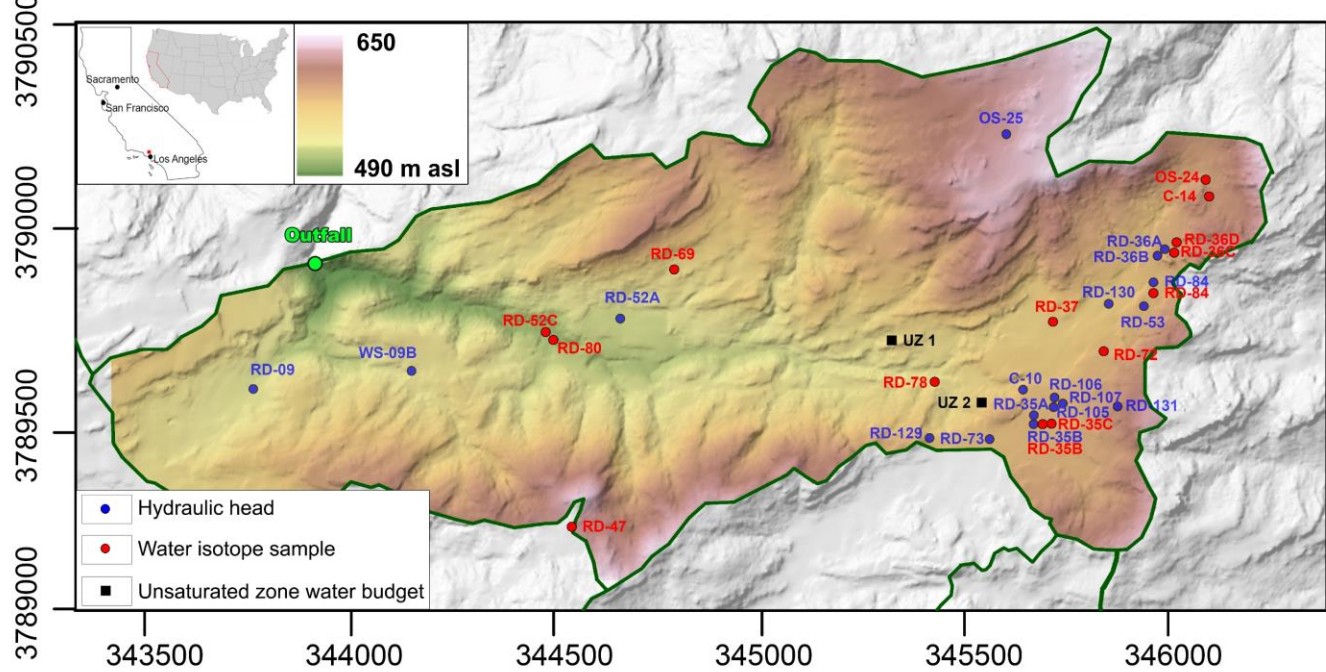

*Figure 1 Topographic map of the study area and location of the wells used for calibration (blue), water isotopes sampling*
*(red). In black the two cells where unsaturated zone water budgets were analyzed.*

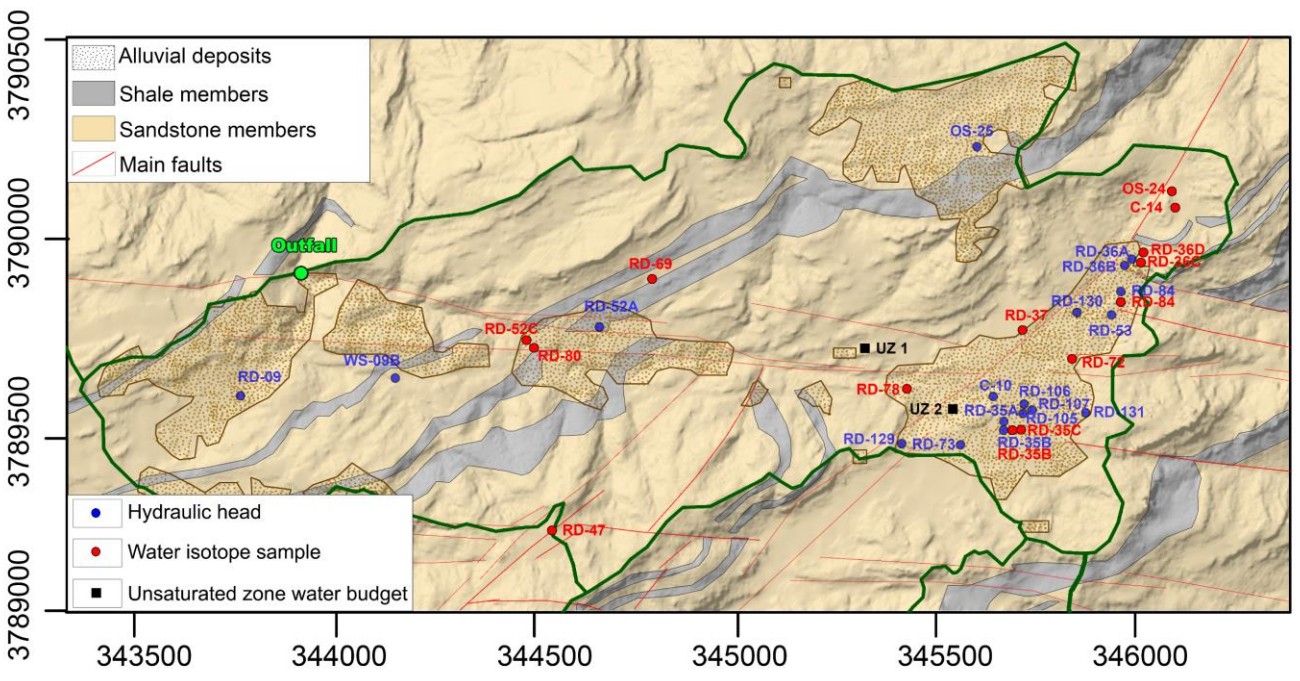


*Figure 2 Geologic map of the study area and location of the wells used for calibration (blue), water isotopes sampling (red).*
*In black the two cells where unsaturated zone water budgets were analyzed.*

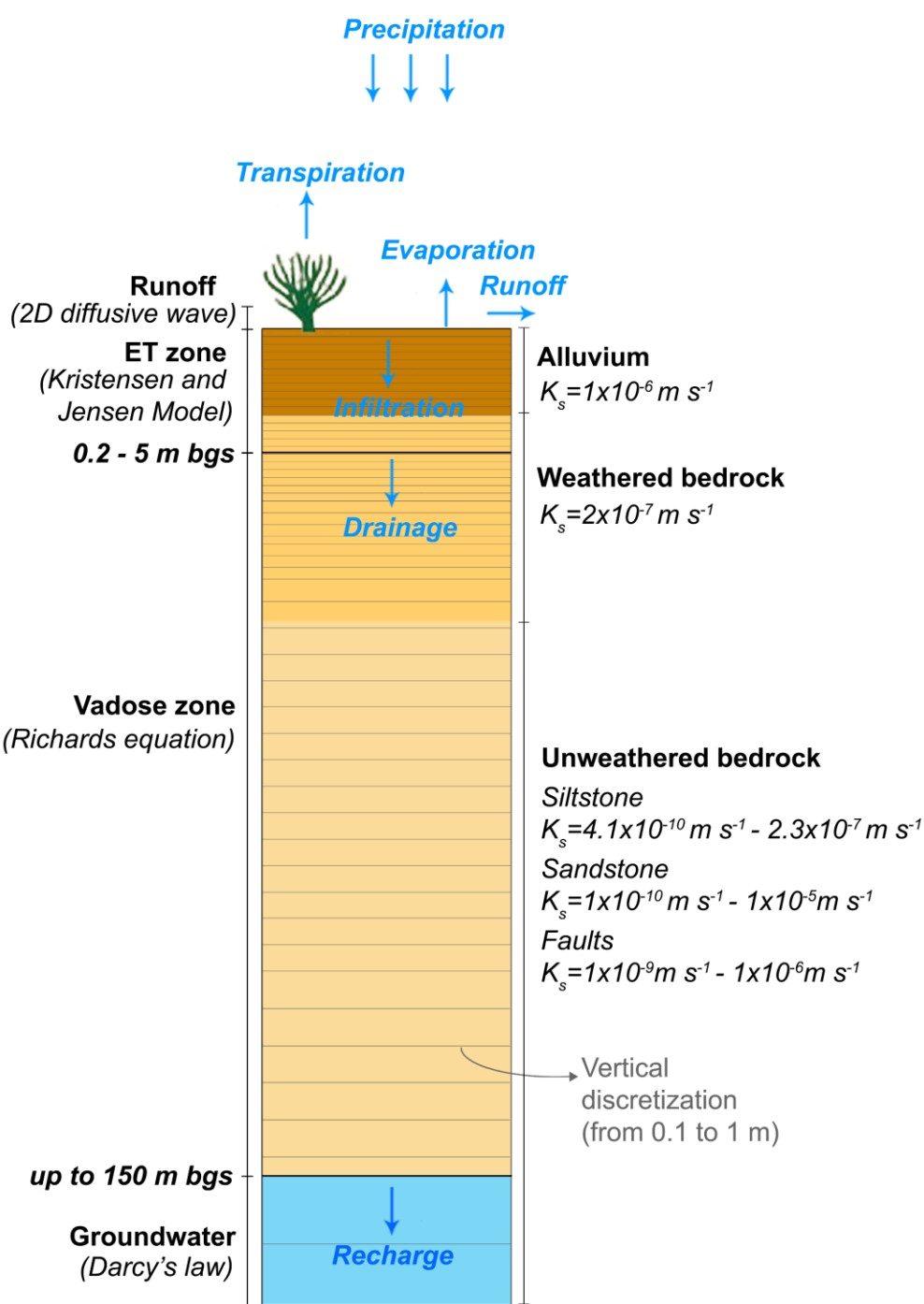

**807**

**808**   *Figure 3 Description of the vertical MIKE SHE model domain*


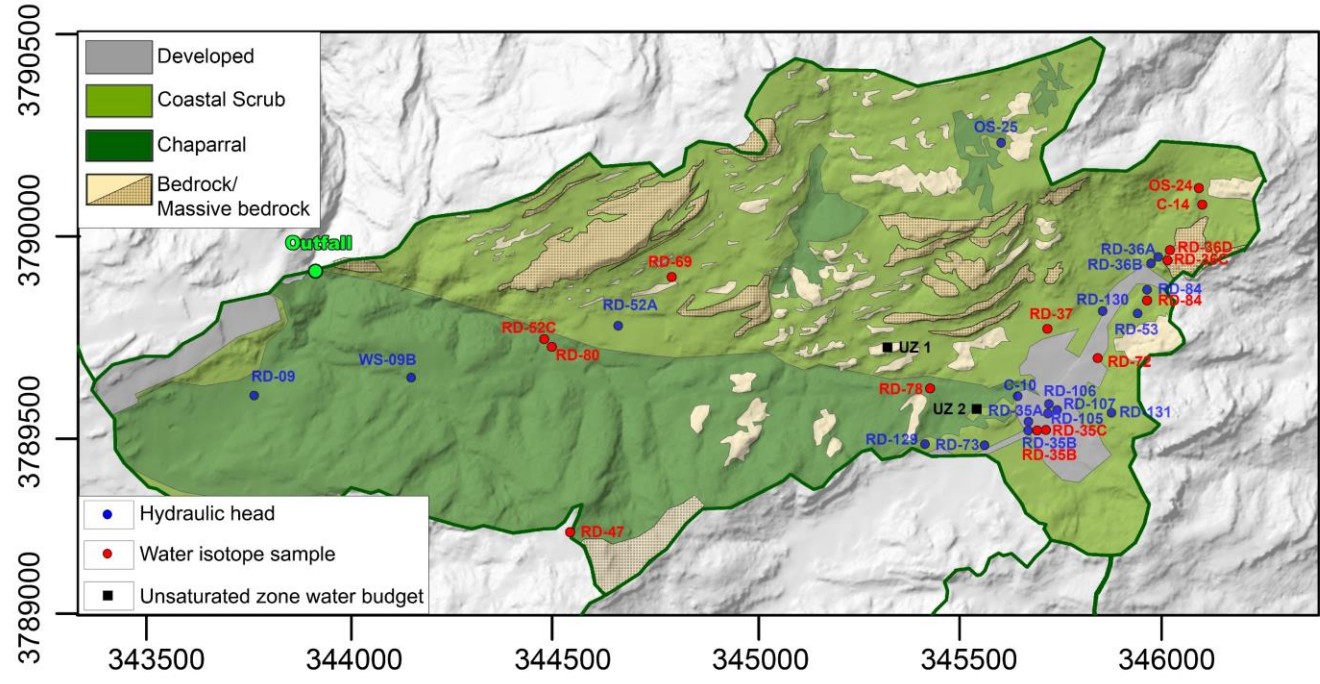

*Figure 4 Land use map and location of the wells used for calibration (blue), water isotopes sampling (red). In black the two*
*cells where unsaturated zone water budgets were analyzed.*

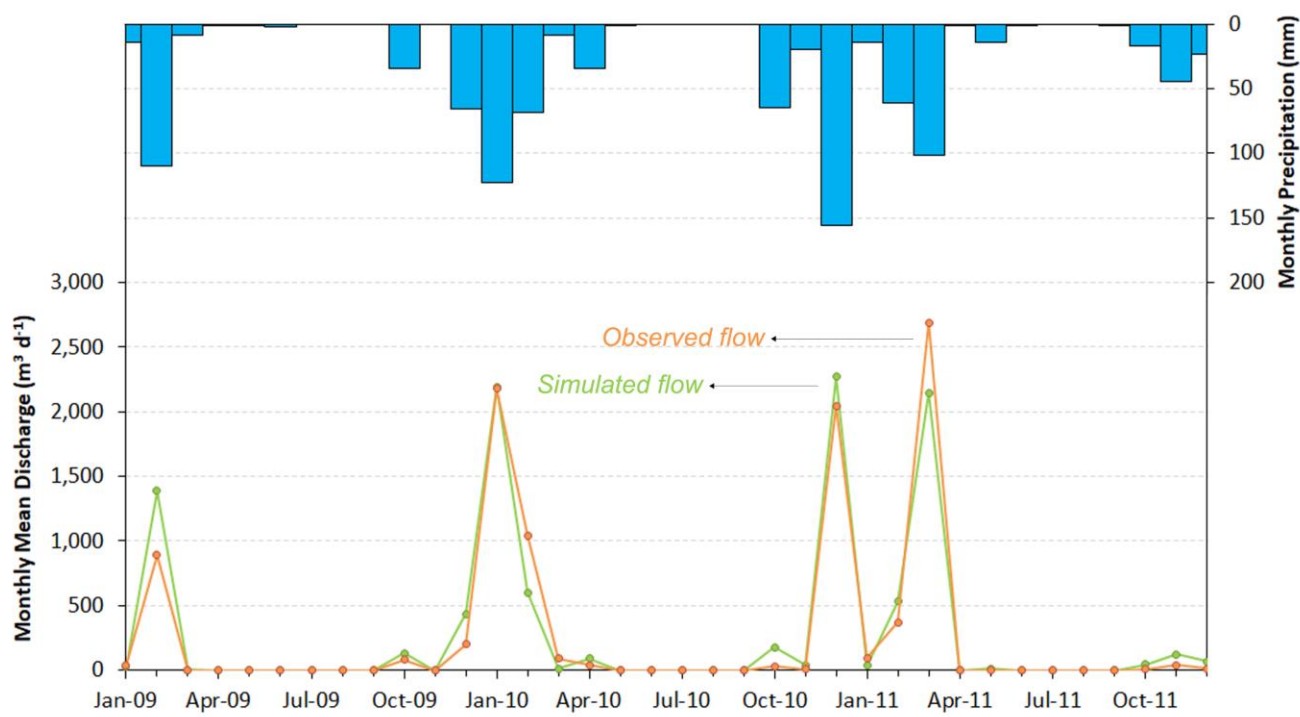


*Figure 5 Monthly precipitation values and comparison between simulated (green) and observed (red) runoff flow at the*
*outfall of the catchment from January 2009 to December 2011.*

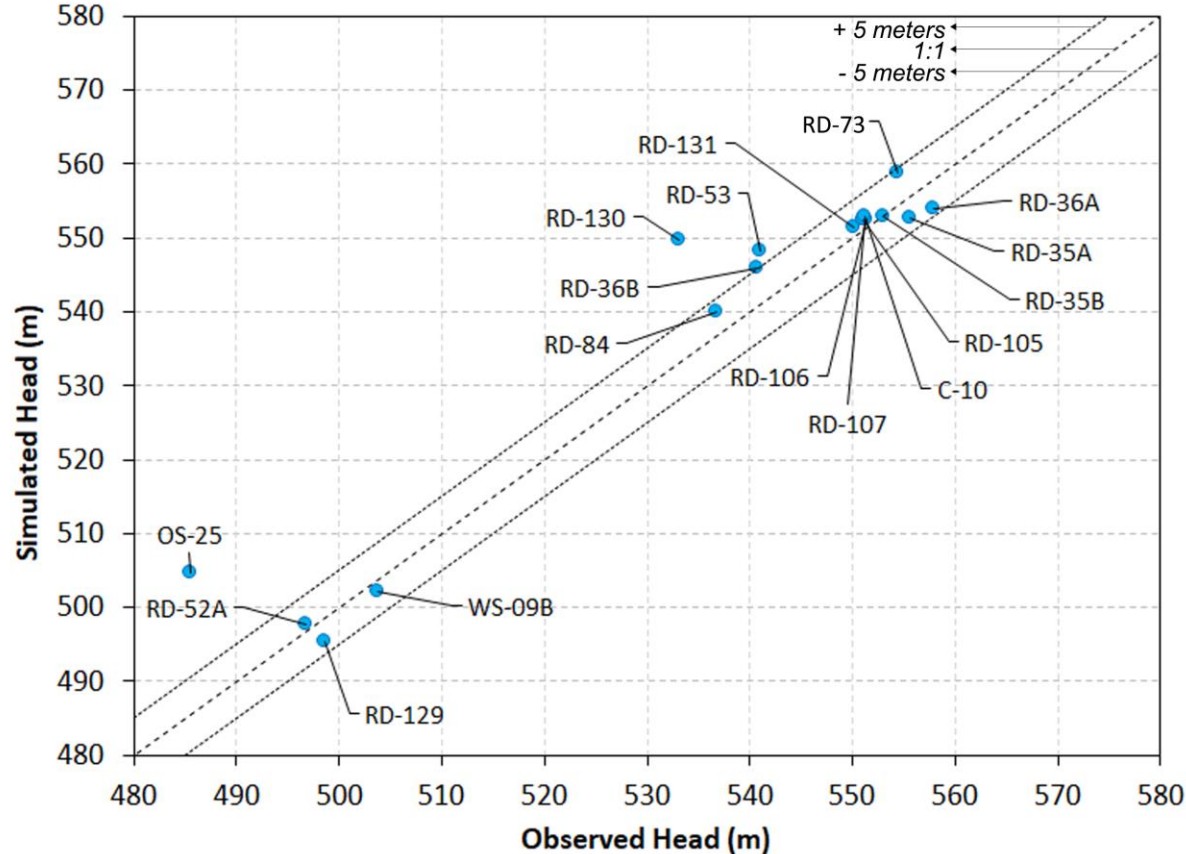


*Figure 6 Comparison between simulated and observed groundwater head data for the 17 wells.*


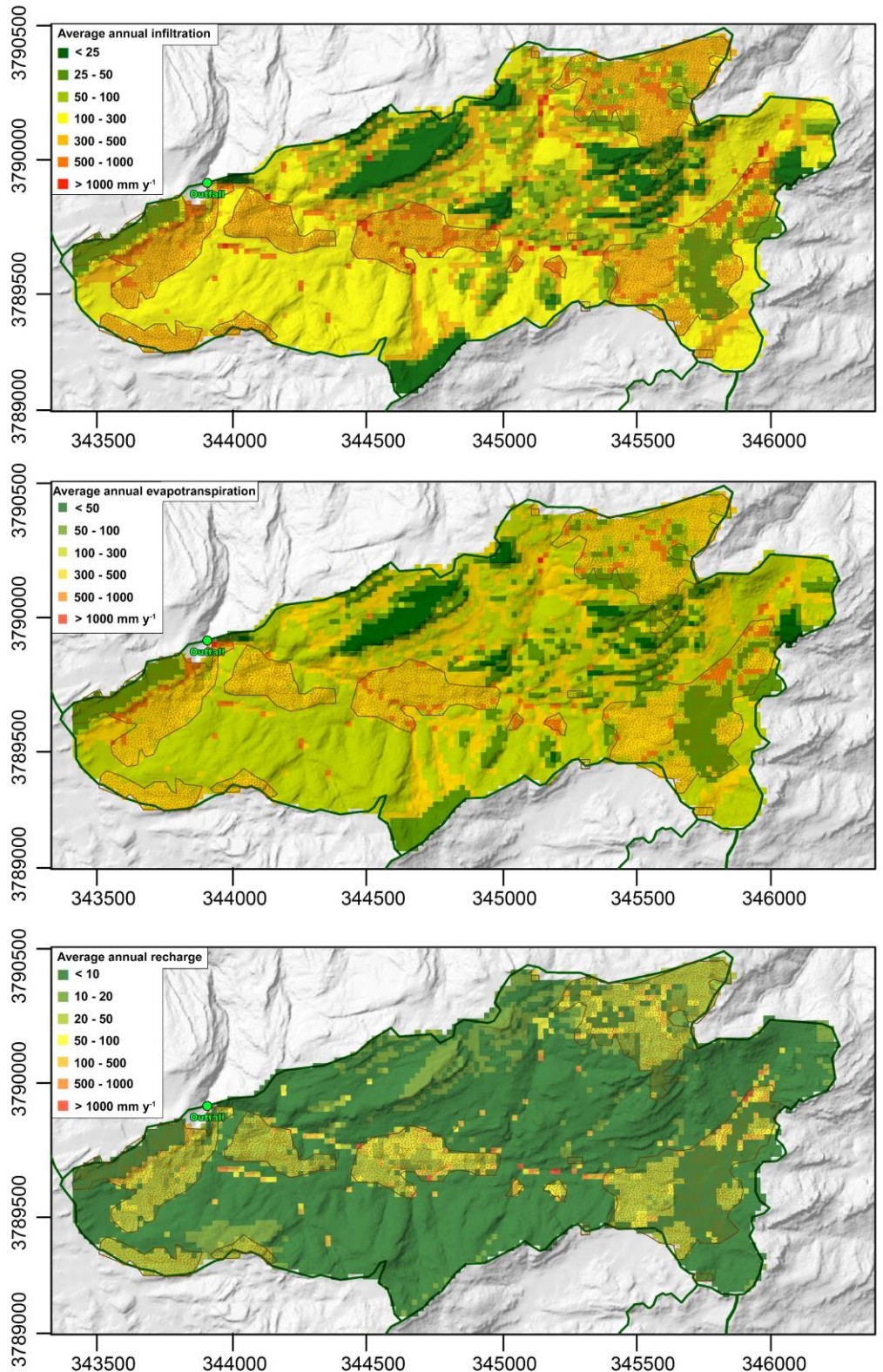


*Figure 7. Distribution of average annual infiltration (a), evapotranspiration (b) and recharge (c). Dashed polygons represent*
*areas with alluvium at the surface.*

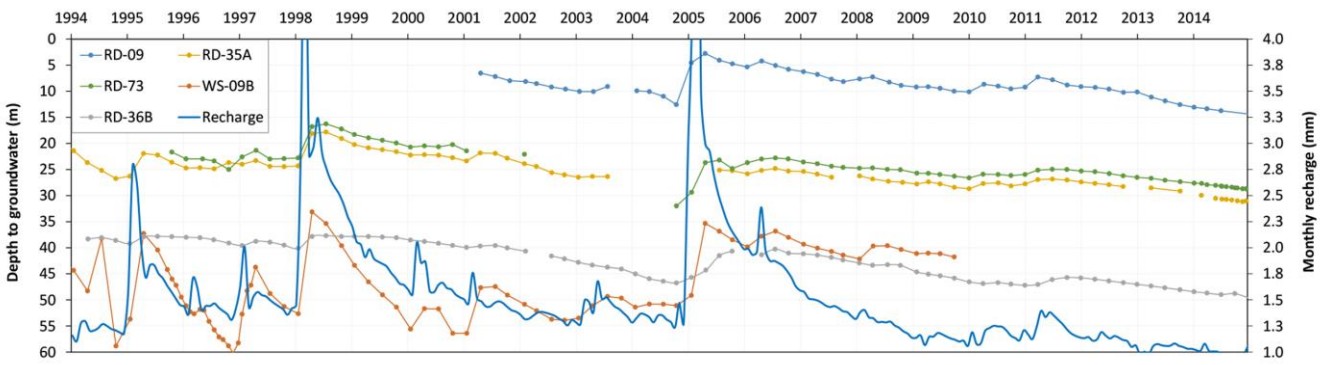

*Figure 8 Unsaturated zone water budget for ET zone from January 2004 to December 2007 for two cells representative of the*
*domain: (a) UZ-1 area with outcropping bedrock without vegetation; (b) UZ-2 area with alluvium deposit covered by*
*vegetation.*


*Figure 9 Comparison between the monthly recharge time series and the depth to groundwater at five locations across the*
*catchment.*

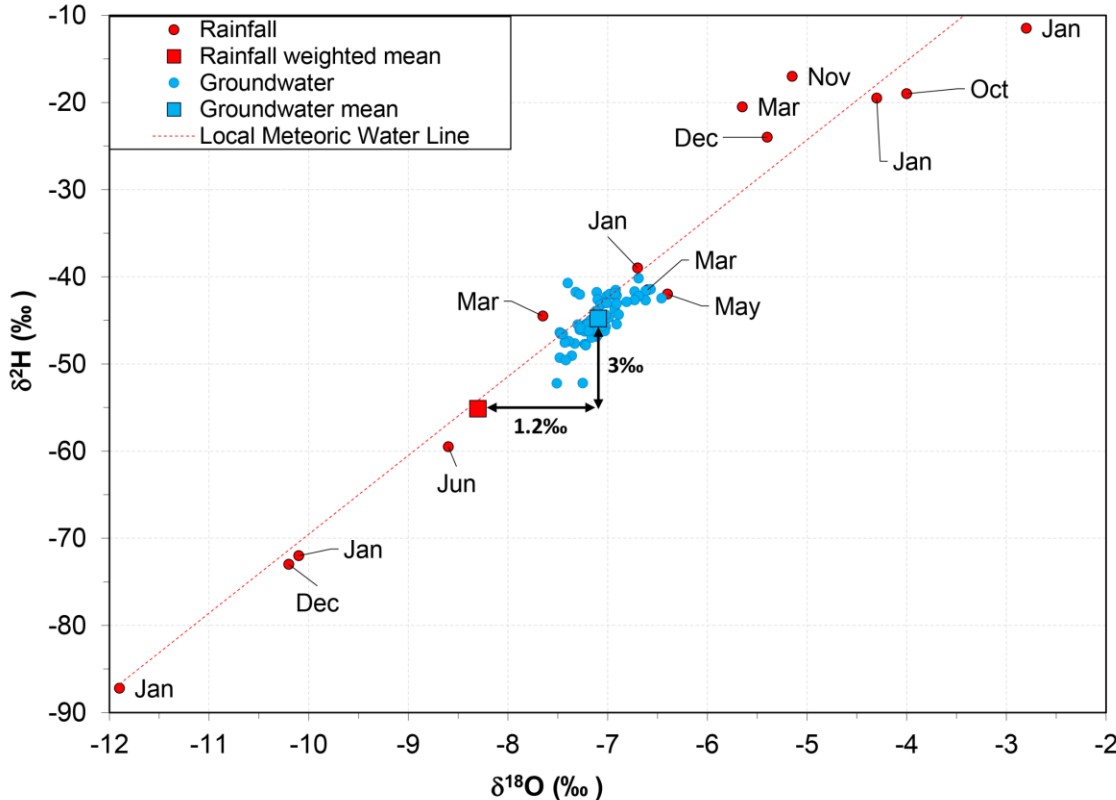


*Figure 10 Water isotopes plot for rainfall samples collected at two rain gauge stations and groundwater samples from 16*
*wells of the catchment.*

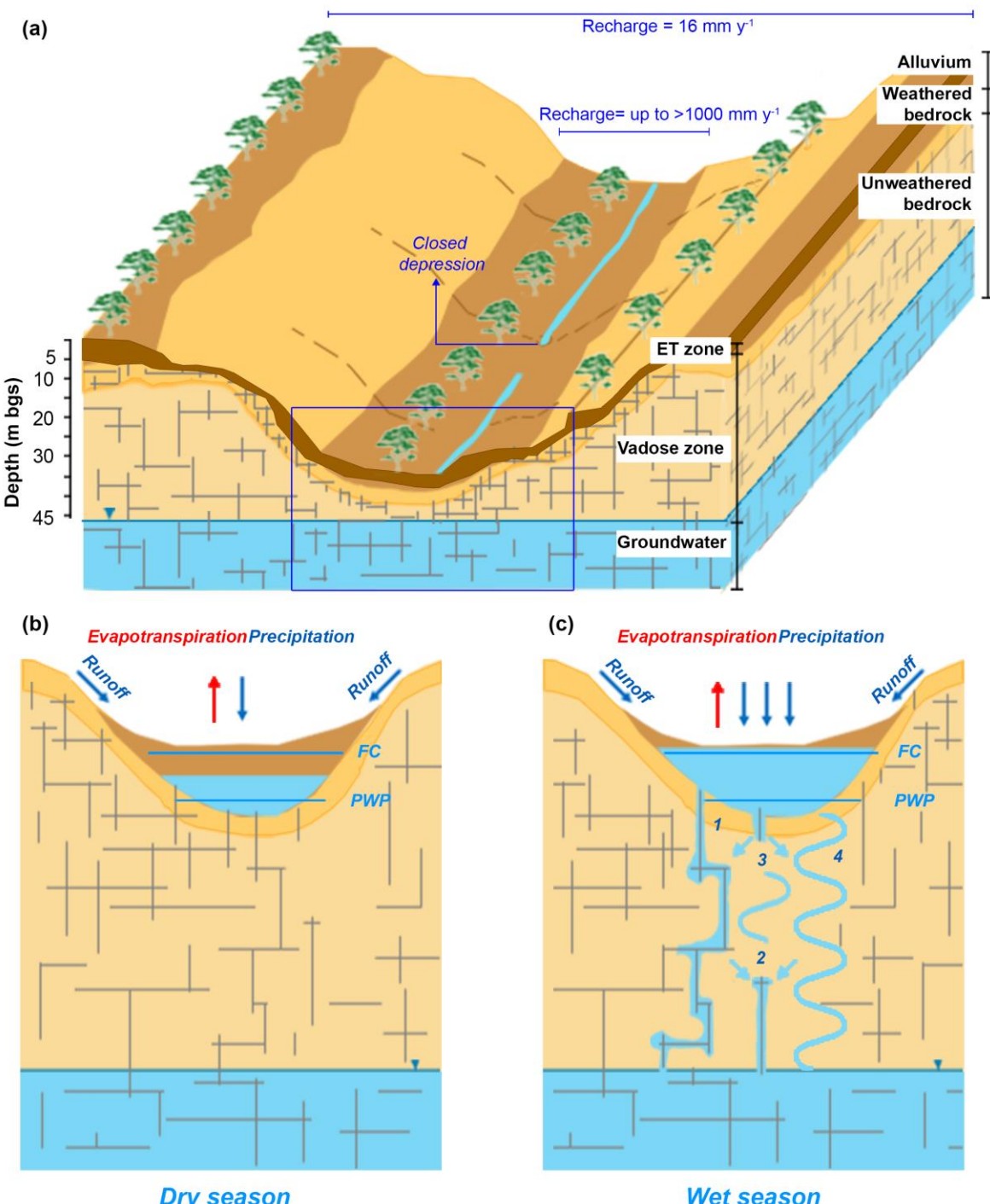


Figure 11 Conceptual model for recharge at the site. (a) Spatial 3-D conceptual model of the catchment showing where high
recharge occurs. 2-D schematic of the unsaturated zone hydrologic process during (b) dry season and (c) wet season. During
the dry season water content is between the field capacity (FC) and the permanent wilting point (PWP) and therefore is
consumed by evapotranspiration. Conversely, during the wet season, water content is above the FC and seeps into the
underlying bedrock. Numbers describe mechanisms of flow in the vadose zone: 1 is fracture flow; 2 is water flowing from
matrix into fractures; 3 is water flux from fractures into matrix; 4 is intergranular matrix flow.

841