# Peer review of "Spatial and temporal variability of groundwater recharge in a sandstone"

_Hydrology and Earth System Sciences, 2018_

## Referee Comment (RC1) · Anonymous Referee #1 · 27 Nov 2018

25-11-2018

A review of: Spatial and temporal variability of groundwater recharge in a sandstone aquifer in a semi-arid region, by Manna et al. submitted to HESS

Summary and Recommendation

In this study, a high-resolution surface-unsaturated_zone-aquifer flow model was fit to a km2 scale hilly drainage basin near Los Angeles, to investigate spatial and temporal variability of groundwater recharge. The main result is that, although the long-term spatial average recharge under the catchment is 16 mm/yr, under the small alluvial valley after heavy rain, focused temporal recharge rate may reach 1000 mm/yr. Although this

type of variability in recharge is not totally new for this setting, the work is worthy for its rare and intensive modelling effort and comparison with local estimates (e.g. chloride mass balance). Nevertheless, substantial changes need to be made in the manuscript before it can be published in HESS.

Major comments

1) Structure: There is no Methods section and no Discussion in the paper. The authors avoiding the classic titles of sections in a scientific paper is deep in the content, many methods are not clear (S. comments 7-10, 13 below), and there is no discussion of the results with the wide literature on recharge. Methods and Discussion sections should be included and taken more seriously (it could be Results and Discussion but a discussion should be done).

2) Concerning the discussion above: I would say that the recharge characteristics described in the manuscript is similar to what many stidies term: Mountain Front Recharge (MFR). Aquifers under alluvial valleys in mountainous regions are recharged from the edge of the valley (mountain front) or maybe altogether in subsurface recharge of rain percolating in the mountain block (can explain fresh groundwater above saline unsaturated zone). Discuss your findings in light of MFR literature.

3) Figures graphics. Although digital era, some of us do print and read from paper some of their work (manuscripts for review, especially). The manuscript include figures with axis-titles that are extremely small (unreadable). Check figures graphics on a printed version with a reader older than 50.

Specific comments

1) L25 The Abstract is a standalone entity, it should not contain references.

2) L49 and throughout the manuscript – put a space after the semicolon.

3) L62 I would change "transient" to fast changing. The literature is full of examples of changing recharge due to change in land-use that were shown via chloride mass

balance and similar methods.

4) L64-L70. In many semiarid regions surface run-off is ~1% of precipitation way within the modeling error, hence sub-surface unsaturated - saturated zone flow models (and in some cases even only unsaturated zone models) are a very reasonable choice for studying recharge and contamination. This type of studies are quite common in the literature of the last decade (e.g. Levi et al., 2017 HESS; Turkeltaub et al., 2015 WRR). Therefore, the elaboration on 2006 review, is outdated and not very convincing, I suggest to discard.

5) L88. Potential evaporation – give the numbers.

6) L 93 chemical contamination – say what contamination (in 2-3 words, nitrate, industrial organic compounds).

7) L140 – How is infiltration capacity modeled? is it constant at field capacity or starts significantly higher after a dry period?

8) L143-146 – Not clear is the root zone and the deeper unsaturated zone modeled as a continuous domain with Richards Equation with root water uptake sink at the root zone. Or is the root-zone modeled as bimodal: above FC –deep drainage, below no deep drainage? "…It is mainly vertical" is it a 1D model in this zone, or of higher dimension.

9) L153-154, as far as I understand if there is a constant head as a bottom boundary condition the water table will not change and recharge or discharge will be reflected only by flux out or into the model domain. Was the model fitted to transient head in wells? or only to a steady-state approximation? If so, say it explicitly in Figure 6 captions.

10) L187 – "physical properties" there is only Ks in the table (not enough to model unsaturated zone flow, parameters of hydraulic functions? What type of functions? – not clear

11) L 242, MIKESHE, MIKE SHE or MIKE-SHE choose 1 and be consistent.

12) L 265, I would change "centuries" to decades in this sentence.

13) L 270-277 when and how these analysis of samples 24 years old were done? Is it new data, if not, reference? If yes a sentence on the analytical methods.

14) L305-307, I assume these are spatially average recharge rates, if right say it explicitly, if not describe.

15) L 449- 452, typical Mountain Front Recharge (major comment 2).

16) L 468 see Kurtzman et al., 2016 HESS, for discussion on by-pass preferential flow recharge of fresh water to aquifers under saline unsaturated zone.

17) Table 3 – rainfall at bottom line is cumulative not mean

18) Figure 1. Confusing map. In physical (topographic) maps green is for low lands and brown for high land. Switch the color scale to fit to the customary color scale.

19) Figure 3 enlarge text

20) Figure 7 enlarge text. m-1 shouldn't be used for per month (its per meter in the SI system).

21) Figure all graphics and writing are too small. Panel C is missing.

Please also note the supplement to this comment:
https://www.hydrol-earth-syst-sci-discuss.net/hess-2018-531/hess-2018-531-RC1-supplement.pdf

---

## Referee Comment (RC2) · Anonymous Referee #2 · 1 Jan 2019

General comments:

The manuscript describes a modeling study of the spatial and temporal variation of recharge in a 2.16 km2 upland catchment in a semi-arid region. Recharge in semi-arid regions constitutes a small fraction of precipitation and is subject to a large temporal and spatial variability. Studies of this hydrological component under semi-arid conditions are relatively few although the references provided by the authors are all more than 10 years old and should thus be updated when revising the manuscript. Nevertheless, I believe that the presented study expands research on recharge in semi-arid regions and that the manuscript deserves publication after revision.

[Figure]

My major concern of the presented work relates to the calibration of the MIKE SHE model, which is inadequately carried out and described. Calibration of a hydrological model should preferably be carried out using an autocalibration method (e.g. PEST) in order to (1) identify the sensitive parameters, (2) calibrate the parameters selected for calibration using an objective method, (3) identify non-uniqueness issues and correlation among the parameters, and (4) identify uncertainty intervals of the calibrated parameter values. The process can be carried out in a more or less sophisticated procedure but in any case it makes the process transparent. The authors do not describe which parameters have been subject to calibration and it is not discussed if the resulting parameters values are reasonable based on prior knowledge of the characteristics of the site. I will encourage the authors to carry out a sensitivity and calibration analysis using an autocalibration method.

My second major concern relates to the conceptualization of the system being studied. The subsurface consists of densely fractured bedrock with parallel beddings and vertical joints and faults leading to preferential flow as also emphasized by the authors at several places in the manuscript. For interpreting chloride and isotope concentration measurements preferential flow appears to be important. Furthermore, the authors have developed a conceptual model for recharge, where distribution between matrix and fractures is described (l. 469-479). The flow processes in and between the two domains are mainly based on speculation and not documented by modelling. The authors need to substantiate why two domains are not considered in their modeling approach.

Specific comments:

l. 66-75: Please update literature review with newer references. l. 103-104: As fracture flow is stated to be an important flow process the authors need to substantiate why this flow process is not considered in the modelling. l. 153-156: Is the lateral boundary condition a closed boundary? Is the lower boundary condition based on field measurements? To which extent will it impact the modeling results? Do I understand correctly

that groundwater does not contribute to stream flow and that all recharge will to deeper aquifer systems? Please elaborate on the model conceptualization. l. 178-179: What are the thicknesses of the two groundwater zone layers? l. 189: Table 2 is incomplete, unsaturated zone characteristics should also be listed. l. 205-211: Could you please be a bit more clear on how the land use are estimated. l. 280- : The calibration procedure needs to be elaborated and revised as described above. l. 301: Generally, I would consider a mean absolute error of 4.5 m to be rather high. Perhaps you mean root mean square error? l. 303-: To me it would make more sense to compare simulated and observed hydraulic heads directly? l. 316- 318: Perhaps the equivalent porous medium approach is suitable for simulation of water flow but for solute transport and the interpretation of chloride and isotopes I am not sure. l. 352: Fig. 8a and 8b. l. 373: Check consistency with lines 216-217.

---

## Author Comment (AC2) · 28 Jan 2019

**Authors' response to Reviewer 2**

**General comment**

The manuscript describes a modeling study of the spatial and temporal variation of recharge in a 2.16 km2 upland catchment in a semi-arid region. Recharge in semi-arid regions constitutes a small fraction of precipitation and is subject to a large temporal and spatial variability. Studies of this hydrological component under semi-arid conditions are relatively few although the references provided by the authors are all more than 10 years old and should thus be updated when revising the manuscript. Nevertheless, I believe that the presented study expands research on recharge in semi-arid regions and that the manuscript deserves publication after revision.

**Author's response:** We thank Reviewer 2 for the thorough review of the paper that highlighted, in particular, a lack of clarity in the calibration section. We think that, replying to the comments, we explained our rationale for the proposed approach.

**Major comments**

**1) Comment from Referee**: My major concern of the presented work relates to the calibration of the MIKE SHE model, which is inadequately carried out and described. Calibration of a hydrological model should preferably be carried out using an autocalibration method (e.g. PEST) in order to (1) identify the sensitive parameters, (2) calibrate the parameters selected for calibration using an objective method, (3) identify non-uniqueness issues and correlation among the parameters, and (4) identify uncertainty intervals of the calibrated parameter values. The process can be carried out in a more or less sophisticated procedure but in any case it makes the process transparent. The authors do not describe which parameters have been subject to calibration and it is not discussed if the resulting parameters values are reasonable based on prior knowledge of the characteristics of the site. I will encourage the authors to carry out a sensitivity and calibration analysis using an autocalibration method.

**Author's response:** Thanks to the reviewer's comment we realized that the text provided in the calibration section was not enough to explain our approach. We will make substantial edits to this section to better describe the rationale and to provide more information about the sensitivity and uncertainty. Here we provide a more in-depth description of our approach.

The parameters involved in the calibration process were surface roughness, detention storage, imperviousness, rooting depth, Leaf Area Index, crop coefficient, hydraulic conductivity and water content parameters of alluvium, hydraulic conductivity and water content parameters of weathered bedrock. We understand that an "autocalibration" approach would have provided additional information and transparency on parameter sensitivity and the uncertainty in recharge estimates. However, we consider our calibration approach to have been rigorous, whereby we tested a range of parameters supported by a large set of field data, against an objective function comprised of groundwater level and stream flow measurements. Results of multiple calibration simulations were

compared against these observations and other data (e.g. chloride mass balance) which helped to further validate the calibration.

The calibration process proceeded in an iterative manner. After each calibration run, the primary calibration parameters were examined with a variety of metrics including:

**Streamflow Calibration Metrics**

- Simulated vs Observed Average Annual flow

    o Mean Error

- Simulated vs Observed Average Monthly Flow:

    o Mean Error

    o Root Mean Squared Error

    o Correlation

    o Nash Sutcliffe Efficiency

- Graphical Plots of Simulated Streamflow Versus Observed Streamflow and Precipitation

    o Provided a qualitative measure of event correlation to observed precipitation and streamflow

**Groundwater Level Calibration Metrics**

- Simulated versus observed water levels

    o Mean Error

    o Mean Absolute Error

    o Root Mean Squared Error

    o Normalized Root Mean Squared Error

- Graphical Plot of Simulated Vs Observed Water Levels (1:1 residual plot)

    o Provided a quantitative and qualitative assessment of the residual error present at observation wells throughout the domain

- Spatial Plot of Groundwater Residuals (map)

    o Provided a quantitative assessment of water level residuals plotted in the model domain

    o Spatial patterns of fit or misfit of the model were compared against other spatial data (e.g. hydraulic conductivity, boundary conditions, land uses, surface geology) to evaluate potential correlations.

Following an assessment of these calibration targets, model parameters were revised to improve representation of the calibration metrics. In instances where the results were not consistent with the site conceptualization consideration was given as to whether an alternative conceptualization would explain the results predicted by the model. From a semi-quantitative assessment of the calibration process, the values of hydraulic conductivity and water content parameters of alluvium and weathered bedrock had the strongest impact on the calibration targets. These deposits represent the upper layers of our model domain and variations in hydraulic conductivity or unsaturated zone properties control the rate of infiltration, evapotranspiration drainage and, therefore, recharge.

The fact that the values are 1) in the same range of those present in literature (Canadell et al., 1996; Scurlock et al., 2001; Chin et al., 2000), 2) similar to those used by the Surface Water Expert Panel (https://www.boeing.com/principles/environment/santa-susana/technical-reports.page) to model surface water flow, 3) in the range of those measured in the groundwater zone during on-site investigations conducted for 20 years (Cherry et al., 2009) helps to significantly constrain the calibration.

**2) Comment from Referee**: My second major concern relates to the conceptualization of the system being studied. The subsurface consists of densely fractured bedrock with parallel beddings and vertical joints and faults leading to preferential flow as also emphasized by the authors at several places in the manuscript. For interpreting chloride and isotope concentration measurements preferential flow appears to be important. Furthermore, the authors have developed a conceptual model for recharge, where distribution between matrix and fractures is described (l. 469-479). The flow processes in and between the two domains are mainly based on speculation and not documented by modelling. The authors need to substantiate why two domains are not considered in their modeling approach.

**Author's response:** Actually, in a previous published paper, the roles of matrix and preferential flow were examined in detail. Analyzing the different average Cl concentration in the vadose zone and in groundwater, Manna et al. (2017) estimated that 80% of the recharge occurs as intergranular flow in the porous matrix block and 20% as fracture flow. Therefore, we think that an EPM model, such as MIKE SHE would reproduce accurately the bulk (matrix -predominantly- and fracture) flow. In addition, the spatial resolution (20 by 20 m cells) is such that the hydrogeological system can be approximated by an EPM model. Our confidence regarding this latter point comes also from the good comparison between the simulated recharge and the observed water level at monitoring wells.

The "conceptual model" section includes findings of previous studies that are incorporated and analyzed in the light of the outcome of the present paper to create indeed a conceptual model. This is why we mention the possible occurrence of preferential flow in the deeper vadose zone, which is not simulated with MIKE SHE but analyzed in previous studies. The text on flow mechanisms at line (469-479) is there to complete the discussion about recharge and unsaturated zone dynamics but not to describe processes that are embedded in our modeling.

**Specific comments**

**1) Comment from Referee**: l. 66-75: Please update literature review with newer references

**Author's response:** We updated the literature following also the suggestions of reviewer 1. However, we want to highlight the surprisingly lack of integrated spatially distributed models for semi-arid catchments in recent years.

**2) Comment from Referee:** l. 103-104: As fracture flow is stated to be an important flow process the authors need to substantiate why this flow process is not considered in the modelling.

**Author's response:** see response to major comment 2.

**3) Comment from Referee:** l. 153-156: Is the lateral boundary condition a closed boundary? Is the lower boundary condition based on field measurements? To which extent will it impact the modeling results? Do I understand correctly that groundwater does not contribute to stream flow and that all recharge will to deeper aquifer systems? Please elaborate on the model conceptualization.

**Author's response:**

There is a fixed head boundary conditions applied to the base and along the lateral faces of the model representing the deep groundwater flow system. The shallow water table and perched systems within the alluvium and weathered bedrock are well above this deeper water table. These heads are based on observed groundwater levels at the site and simulations based on a detailed groundwater flow model. Given that the groundwater heads associated with deep aquifer system are generally observed at relatively large depths below ground surface throughout the domain, it is expected that variations in these specific values assigned would not have a significant effect on predicted recharge values. In areas where the groundwater is observed to be closer to ground surface, the alteration of these values could potentially have a more direct effect on groundwater recharge in that a groundwater table close to the surface could rise to meet the ground surface given sufficient recharge.

It is correct that groundwater contribution to streamflow is minimal and only occurs after rainfall event at the farthest downstream location of the catchment. At the outlet of the catchment where the groundwater table rises close to the ground surface and there is intermittent and limited (~ 0.1 mm $y^{-1}$ for the period of 1995-2014) contribution of groundwater to streamflow leaving the system.

**4) Comment from Referee:** l. 178-179: What are the thicknesses of the two groundwater zone layers?

**Author's response:** Layer 1 has a thickness variable from 24 to 185 m (average: 109 m) whereas layer has a uniform thickness of 5 m. While layer 1 may appear very thick the 'active' part from a numerical perspective begin only when the water table is reached. Flow above that occurs in the unsaturated zone that features a finer discretization.

**5) Comment from Referee:** l. 189: Table 2 is incomplete, unsaturated zone characteristics should also be listed.

**Author's response:** The table has been completed with porosity, field capacity, residual water content and the Van Genuchten parameters (α, n) used in the model.

The model uses three separate sets of Van Genuchten parameter to represent the pressure-saturation-hydraulic conductivity relationships; 1) alluvium, 2) weathered bedrock, 3) un-weathered bedrock. The parameters used reflect our understanding that the rock matrix transmits the largest volume of recharge, while recharge through the fractures is faster. The relationships used are biased towards the matrix response. These values were further calibrated using the groundwater level responses and the stream flow. Further rock core samples indicate a high moisture content (~80%) indicating that K is often close to $K_s$ and the hydraulic conductivity-saturation curve reflects this understanding.

| Hydrogeologic unit | $K_s$ (m s$^{-1}$) | Saturation ($\theta_s$) | Field capacity ($\theta_{fc}$) | Residual Water content ($\theta_{fc}$) | Van Genuchten parameters | | |
|---|---|---|---|---|---|---|---|
| | | | | | $\alpha$ | n | l |
| Alluvium | $1\times10^{-6}$ | 0.4 | 0.25 | 0.05 | 0.021 | 1.61 | 0.5 |
| Weathered bedrock | $2\times10^{-7}$ | 0.2 | 0.11 | 0.01 | 0.033 | 1.49 | 0.5 |
| Unweathered bedrock | $4.1\times10^{-10}$ to $2.3\times10^{-7}$ | 0.13 | 0.1 | 0.025 | 0.01 | 1.23 | 0.5 |
| Unweathered bedrock | $1\times10^{-10}$ to $1\times10^{-5}$ | 0.13 | 0.09 | 0.01 | 0.01 | 2 | 0.5 |
| Unweathered bedrock | $1\times10^{-9}$ to $1\times10^{-6}$ | 0.13 | 0.1 | 0.025 | 0.01 | 2 | 0.5 |

**6) Comment from Referee:** l. 205-211: Could you please be a bit more clear on how the land use are estimated.

**Author's response:** Land use classes were identified and delineated based on aerial imagery and local land cover datasets (Davis et al., 1998). Descriptions of vegetation classes and species were used in conjunction with literature values for vegetation rooting depth and leaf area indices to describe local vegetation within the model.

**7) Comment from Referee:** L140 – l. 280- : The calibration procedure needs to be elaborated and revised as described above.

**Author's response:** see main comment 1.

**8) Comment from Referee:** l. 301: Generally, I would consider a mean absolute error of 4.5 m to be rather high. Perhaps you mean root mean square error?

**Author's response:** We agree that 4.5 might be seen as high error. However, we are in a recharge area, on a topographic high with hundreds of meters of head potential. In addition, given the complex structural setting (faults located in the deeper system -not modeled), the heterogeneity of the media (porosity ranging between 2 and 20% within meters observed in rock cores, hydraulic conductivities between $1\times10^{-5}$ and $1\times10^{-10}$ m s$^{-1}$) and the vertical discretization of the model around the water table, we think that 4.5 m is a reasonable mean error.

**9) Comment from Referee:** l. 303-: To me it would make more sense to compare simulated and observed hydraulic heads directly?

**Author's response:** Unfortunately, given the subsurface heterogeneity (see response to comment 8) and the spatial variability of recharge is difficult to reproduce reliable transient head time series. The goal of showing this comparison is to validate the ability of the model to reproduce transient recharge conditions over the studied catchment and we think that the good match between simulated recharge and observed water level provides this confidence.

**10) Comment from Referee:** l. 316- 318: Perhaps the equivalent porous medium approach is suitable for simulation of water flow but for solute transport and the interpretation of chloride and isotopes I am not sure.

**Author's response:** Agree but this is truer for the saturated zone than for the vadose zone. As explained in the response to the main comment 2, a previous study found that at the site recharge (and Cl transport) occurs mainly as intergranular matrix flow in the vadose zone. Therefore, we think that our EPM model can be corroborated by recharge studies based on the Chloride Mass Balance method and that the isotopic composition of groundwater can be interpreted under an EPM conceptual model (especially because the Et zone is made of alluvium and weathered bedrock).

**11) Comment from Referee**: l. 352: Fig. 8a and 8b.

**Author's response:** Ops! We replaced 7b with 8b.

**12) Comment from Referee:** l. 373: Check consistency with lines 216-217.

**Author's response: Thanks.** We made it consistent.

---

## Author Response (AR1)

**Authors' response to Reviewer 1**

**Summary and Recommendation**

**Comment from Referee:** In this study, a high-resolution surface-unsaturated zone-aquifer flow model was fit to a km2 scale hilly drainage basin near Los Angeles, to investigate spatial and temporal variability of groundwater recharge. The main result is that, although the long-term spatial average recharge under the catchment is 16 mm/yr, under the small alluvial valley after heavy rain, focused temporal recharge rate may reach 1000 mm/yr.
Although this type of variability in recharge is not totally new for this setting, the work is worthy for its rare and intensive modelling effort and comparison with local estimates (e.g. chloride mass balance). Nevertheless, substantial changes need to be made in the manuscript before it can be published in HESS.

**Author's response:** We thank reviewer 1 for the positive feedback and for recognizing the modeling effort we put in place. We tried to respond exhaustively to all the comments and modify the text accordingly.

**Major comments**

**1) Comment from Referee**: Structure: There is no Methods section and no Discussion in the paper. The authors avoiding the classic titles of sections in a scientific paper is deep in the content, many methods are not clear (S. comments 7-10, 13 below), and there is no discussion of the results with the wide literature on recharge. Methods and Discussion sections should be included and taken more seriously (it could be Results and Discussion but a discussion should be done).

**Author's response:** The description of the methodology used is in the MIKE SHE model section. We expanded this section to make it clearer and more comprehensive, responding to the reviewers' comments.
A discussion about recharge characteristics and about the occurrence of preferential flow in the ET zone has been added to the "conceptual model of recharge" that now has become "Discussion and conceptual model for recharge" section.

**Author's changes in manuscript**:

Comments 7-10 and 13 were addressed (see response to specific comments below for details about changes in the manuscript).

Text added to the discussion section:

- Line 520 – 523: The average recharge value is 16 mm y$^{-1}$ which is consistent with previous estimates at the site, and with those obtained for other sandstone aquifers in semi-arid areas in the United States (4% - Heilweil et al., 2006) and other studies in semi-arid regions around the world (0.2 – 35 mm y$^{-1}$ equal to 0 – 5% of the average precipitation, Scanlon et al., 2006).

- Line 529-535: Generally, in semiarid regions, high recharge values along a valley, at the edge of the slope referred to as Mountain Front Recharge (MFR) (Wilson and Guan, 2004). However, our catchment is located on the top of a ridge standing 300 m above the surrounding valleys (Manna et al., 2016) and, thus, our case study represents groundwater recharge on the mountain block rather than MFR. Nonetheless, it is interesting that the processes observed in our small catchment are similar to those described for aquifer-scale recharge studies (Aishlin and McNamara, 2011; Carling et al., 2012; Manning and Solomon, 2003; Bresciani et al., 2018) and defined as MFR.

- Line 550-556: Case studies showing similar results for water that crosses the ET zone preferentially in time and space to become potentially recharge have been also reported in literature (Kurtzman et al., 2016), also referred to as selective recharge (Gat and Tzur, 1967; Florea, 2013; Krabbenhoft et al., 1990) . The occurrence of these fluxes has been also analyzed in function of precipitation characteristics and antecedent water content with rainfall intensity being the main factor (Allocca et al., 2015; Crosbie et al., 2012; Nasta et al., 2018; Taylor et al., 2013).

**2) Comment from Referee**: Concerning the discussion above: I would say that the recharge characteristics described in the manuscript is similar to what many studies term: Mountain Front Recharge (MFR). Aquifers under alluvial valleys in mountainous regions are recharged from the edge of the valley (mountain front) or maybe altogether in subsurface recharge of rain percolating in the mountain block (can explain fresh groundwater above saline unsaturated zone). Discuss your findings in light of MFR literature.

**Author's response:** We thank you the reviewer for this suggestion that allowed us to describe better our conceptual model and the hydrologic processes involved. The spatial distribution of recharge and the proposed conceptual model might recall what has been defined Mountain Front Recharge (Wilson and Guan, 2004). However, the catchment is located on an upland ridge that represents, on a regional scale, the mountain block. Although the processes observed are similar to those described as diffuse and focused MFR (direct water-table recharge at the edge of a slope front), we believe that recharge characteristics are more similar to recharge at the mountain block. A classic MFR and MBR approach would have been more plausible at regional scale, perhaps including the surrounding Simi and San Fernando valleys (about 300 m below the studied catchment). Instead, we only focused on a small watershed (2.16 km$^2$), with a local relief of 150 m located on the top of the Simi Hills. The maximum thickness of the alluvium overlying the sandstone bedrock in the low areas of the catchment, where the majority of recharge occurs, is only 3.8 m and therefore this setting is different from the alluvial-filled basins described in several Mountain Front recharge papers (Aishlin and McNamara, 2011; Carling et al., 2012; Manning and Solomon, 2003; Bresciani et al., 2018). Given all these considerations, we believe that the system represents a small portion of the Mountain Block rather than the MFR (see Figure 1 at the end of the document) although, by analogy to much different scale, we add reference to these concepts in our discussion.

The presence of less saline groundwater below a more saline vadose zone in our case has been attributed by Manna et al. (2017) to preferential flow along the fracture network in the vadose zone. This fast component of the unsaturated flow represents, on average, only 20% of the total recharge with the majority of the flow occurring in the porous matrix blocks.

**Author's changes in manuscript:** We added some text with reference to MFR in the discussion and conceptual model (see comment 1)

**3) Comment from Referee**: Figures graphics. Although digital era, some of us do print and read from paper some of their work (manuscripts for review, especially). The manuscript include figures with axis-titles that are extremely small (unreadable). Check figures graphics on a printed version with a reader older than 50.

**Author's response:** We increased the size of the fonts to improve the readability.

**Specific comments**

**1) Comment from Referee**:  L25 The Abstract is a standalone entity, it should not contain references.

**Author's response:** Accepted. We removed references from the abstract

**2) Comment from Referee:**  L49 and throughout the manuscript – put a space after the semicolon.

**Author's response:** Accepted. We modified throughout the manuscript.

**3) Comment from Referee:**  L62 I would change "transient" to fast changing. The literature is full of examples of changing recharge due to change in land-use that were shown via chloride mass balance and similar methods.

**Author's response:** We changed to "dynamic, short-term" temporal effects

**4) Comment from Referee:**  L64-L70. In many semiarid regions surface run-off is ~1% of precipitation way within the modeling error, hence sub-surface unsaturated - saturated zone flow models (and in some cases even only unsaturated zone models) are a very reasonable choice for studying recharge and contamination. This type of studies are quite common in the literature of the last decade (e.g. Levi et al.,

HESS; Turkeltaub et al., 2015 WRR). Therefore, the elaboration on 2006 review, is outdated and not very convincing, I suggest to discard.

**Author's response:**
Embracing the reviewer's suggestion, we added more recent references of recharge studies in semi-arid environments using different approaches. The elaboration on Scanlon et al., 2006 was introduced to show that until that date only few modeling studies were carried out in semiarid regions, mainly at the regional scale. We left the reference to the main paper and discarded the citations of the single studies. Anyway, we would like to highlight the lack of papers that feature an integrated surface water and groundwater approach in semiarid environments. Sometime, as the reviewer pointed out, this interaction can be considered negligible but, in several cases (like the presented manuscript), it has a huge impact on the spatial distribution of recharge.

**Author's changes in manuscript**: line 64-76. Text added.
Numerical hydrologic models that integrate surface water and groundwater flows have been developed to simulate the spatial and temporal distribution of surface runoff, infiltration, evapotranspiration and groundwater recharge. However, the application of nearly all such simulation tools have been limited to humid regions (Wheater et al., 2007) with minimal application to semiarid regions. Scanlon et al. (2006), in their review on recharge in semiarid areas, reported only 7 papers providing a continuous spatial distribution of recharge, out of a total of 98 studies. However, these studies investigated large areas, from 1,039,647 km$^2$ (Flint and Flint, 2007) to 60 km$^2$ (Flint et al., 2001), using a relatively coarse spatial resolution (from 72,900 m$^2$ - Flint and Flint, 2007 to 900 m$^2$ - Flint et al., 2001).  In the last decade, although modeling techniques have advanced to include combined surface water-groundwater simulations, recharge in semiarid areas has been represented with a GIS approach (Hernández-Marín et al., 2018) often using remote sensing data (Wang et al., 2008; Coelho et al., 2017; Crosbie et al., 2015) or neglecting the surface water component and focusing on unsaturated zone  (Levy et al., 2017; Turkeltaub et al., 2015).

**5) Comment from Referee:**  L88. Potential evaporation – give the numbers.

**Author's response:** 1400 mm y$^{-1}$. Added to the text.

**6) Comment from Referee:**  L 93 chemical contamination – say what contamination (in 2-3 words, nitrate, industrial organic compounds).

**Author's response:** The main contaminant is Trichloroethene (TCE). Added to the text.

**7) Comment from Referee:**  L140 – How is infiltration capacity modeled? is it constant at field capacity or starts significantly higher after a dry period?

**Author's response:** Infiltration capacity of soil in the model is dynamic and a function of the conductivity of the surficial material and the water content properties (saturation point, field capacity and wilting point). The conductivity of the soils is a function of degree of saturation in the soil and a soil moisture characteristic curves. The soil moisture characteristic curve describes the variation in soil water content and conductivity and matric potential. The Van Genuchten model is used to describe the soil moisture characteristic curves in this MIKE SHE model. The conductivity and matric potential of subsurface materials is computed for each layer within the unsaturated zone at each time step. Values used have been added to table 2 for more clarity.

**Author's changes in manuscript:** line 148-152. Text added: The infiltration capacity in the model is dynamic and a function of the unsaturated hydraulic conductivity ($K_u$) and the water content properties (i.e., saturation point, field capacity and permanent wilting point) of the surficial media. To describe the relation between water content, conductivity and matric potential, the Van Genuchten model is used (Van Genuchten, 1980)

**8) Comment from Referee:** L143-146 – Not clear is the root zone and the deeper unsaturated zone modeled as a continuous domain with Richards Equation with root water uptake sink at the root zone. Or is the root-zone modeled as bimodal: above FC –deep drainage, below no deep drainage?

"…It is mainly vertical" is it a 1D model in this zone, or of higher dimension.

**Author's response:** The unsaturated zone is a continuous domain that is modelled as a 1D column of finite difference cells which have variable discretization from the top of the column (ground surface) to the base of the column (the unsaturated/saturated zone interface). The Richard's equation governs flow throughout the unsaturated zone. Typically, when we refer to the root zone we are describing that portion of the unsaturated zone in which vegetation has roots and the capillary fringe which may exist below the roots themselves.

**Author's changes in manuscript:** Line 155-159. Text added: The unsaturated zone flow is simulated as the change in soil moisture, resulting from cyclical input (infiltration) and output (recharge and evapotranspiration). It is modelled as a 1D column using the full Richards equations (Richards, 1931) with finite difference cells that have variable discretization from the top of the column (ground surface) to the base of the column (the unsaturated/saturated zone interface).

**9) Comment from Referee:** L153-154, as far as I understand if there is a constant head as a bottom boundary condition the water table will not change and recharge or discharge will be reflected only by flux out or into the model domain. Was the model fitted to transient head in wells? or only to a steady-state approximation? If so, say it explicitly in Figure 6 captions.

**Author's response:** There is a fixed head boundary conditions applied to the base of the model based on observed groundwater levels. If heads in the layer above the base layer of the model exceed the fixed heads then water will flow out of the model, conversely if heads in the layer above the base layer of the model fall below those in the fixed head then water will flow into the model. The model was calibrated to long term average groundwater levels over the period of simulation (1995-2014).

**Author's changes in manuscript.** Line 166 – 174. Text added: A fixed head boundary applied along the lateral sides and the bottom of the model domain (490 m asl) was used to simulate the flow to and from the deeper groundwater system, not explicitly represented in the integrated model but which extends several hundred meters (Fig. 3). These fixed heads are based on observed groundwater levels at the site and simulations based on a detailed 3-D groundwater flow model system that includes the catchment and a much larger domain beyond (AquaResource and MWH, 2007). The groundwater contribution to streamflow is minimal and intermittent (~ 0.1 mm y$^{-1}$ for the period of 1995-2014) and only occurs at the farthest downstream location of the catchment where the groundwater table rises close to the ground surface.

**10) Comment from Referee:** L187 – "physical properties" there is only Ks in the table (not enough to model unsaturated zone flow, parameters of hydraulic functions? What type of functions? – not clear

**Author's response:** The table has been completed with porosity, field capacity, residual water content and the Van Genuchten parameters (α, n) used in the model.

| Hydrogeologic unit | K$_s$ (m s$^{-1}$) | Saturation (θ$_s$) | Field capacity (θ$_{fc}$) | Residual Water content (θ$_{fc}$) | Van Genuchten parameters | | |
|---|---|---|---|---|---|---|---|
| | | | | | α | n | l |
| Alluvium | 1×10$^{-6}$ | 0.4 | 0.25 | 0.05 | 0.021 | 1.61 | 0.5 |
| Weathered bedrock | 2×10$^{-7}$ | 0.2 | 0.11 | 0.01 | 0.033 | 1.49 | 0.5 |
| Unweathered bedrock | 4.1×10$^{-10}$ to 2.3×10$^{-7}$ | 0.13 | 0.1 | 0.025 | 0.01 | 1.23 | 0.5 |
| Unweathered bedrock | 1×10$^{-10}$ to 1×10$^{-5}$ | 0.13 | 0.09 | 0.01 | 0.01 | 2 | 0.5 |
| Unweathered bedrock | 1×10$^{-9}$ to 1×10$^{-6}$ | 0.13 | 0.1 | 0.025 | 0.01 | 2 | 0.5 |

The model uses three separate sets of Van Genuchten parameter to represent the pressure-saturation-hydraulic conductivity relationships; 1) alluvium, 2) weathered bedrock, 3) un-weathered bedrock. The parameters used reflect our understanding that the rock matrix transmits the largest volume of recharge, while recharge through the fractures is faster. The relationships used are biased towards the matrix response. These values were further calibrated using the groundwater level responses and the stream flow. Further rock core samples indicate a high moisture content (~80%) indicating that K is often close to K$_s$ and the hydraulic conductivity-saturation curve reflects this understanding.

**Author's changes in manuscript:** line 204-215. Text added: The surface and subsurface hydrogeologic units include alluvium, fractured weathered and unweathered bedrock comprised of sandstone, siltstone and shale beds of varying thickness, grain size and cementation (Fig. 2 and Fig. 3). The physical properties of these units, derived from previous on-site investigations (Allegre et al., 2016; Quinn et al., 2015; Quinn et al., 2016) and adjusted by calibration, are summarized in Table 2. In particular, our model uses three separate sets of Van Genuchten parameters to represent the pressure saturation-hydraulic conductivity relationships. The parameters used reflect our understanding that the rock matrix transmits the largest volume of recharge (80%), while recharge through the fractures is minimal (20%) (Manna et al., 2017). Therefore, the relationships used are biased towards the matrix response. These values were further calibrated using the groundwater level responses and the streamflow. Further rock core samples indicate a high moisture content (~80%) (Cherry et al., 2009) indicating that $K_u$ is often close to $K_s$ and the hydraulic conductivity-saturation curve reflects this understanding.

**11) Comment from Referee:** L 242, MIKESHE, MIKE SHE or MIKE-SHE choose 1 and be consistent.

**Author's response:** It is MIKE SHE. We made it consistent throughout the text.

**12) Comment from Referee:** L 265, I would change "centuries" to decades in this sentence.

**Author's response:** Changed to decades.

**13) Comment from Referee:** L 270-277 when and how these analysis of samples 24 years old were done? Is it new data, if not, reference? If yes a sentence on the analytical methods.

**Author's response:** Oxygen isotope ($^{18}O/^{16}O$) and hydrogen isotope ($^2H/^1H$) ratios were measured on an automated gas-source mass spectrometer at the Center for Isotope Geochemistry at the University of California Berkeley laboratory. Water samples for O-isotope analysis were inlet directly into an automated, computer driven gas equilibration system attached to the mass spectrometer. Hydrogen gas samples were prepared for D/H ratio analysis using conventional reduction methods over heated zinc beads in closed tubes. The hydrogen gas was inlet to the mass spectrometer through an automated inlet system.

**Author's changes in manuscript:** L329-332. Text added: The available isotope data for rainfall were determined for the period October 1994 to June 1995 collected at two rain gauge stations (B/886 and RMDF), 5 km from the studied watershed and analyzed in the same year by an automated gas-source mass spectrometer at the University of California Berkeley.

**14) Comment from Referee:** L305-307, I assume these are spatially average recharge rates, if right say it explicitly, if not describe.

**Author's response:** Correct.

**Author's changes in manuscript:** This portion of the text was moved to the Model validation (line 456). We added "spatial average" to line 459.

**15) Comment from Referee:** L 449- 452, typical Mountain Front Recharge (major comment 2).

**Author's response:** see response to major comment 2

**16) Comment from Referee:** L 468 see Kurtzman et al., 2016 HESS, for discussion on by-pass preferential flow recharge of fresh water to aquifers under saline unsaturated zone.

**Author's response:** We added a reference to Kurtzmann et al., 2016 and we also added references regarding the link between precipitation characteristics and preferential flow.

**Author's changes in manuscript:** Line 550-556: Case studies showing similar results for water that crosses the ET zone preferentially in time and space to become potentially recharge have been also reported in literature (Kurtzman et al., 2016), also referred to as selective recharge (Gat and Tzur, 1967; Florea, 2013; Krabbenhoft et al., 1990) . The occurrence of these fluxes has been also analyzed in function of precipitation characteristics and antecedent water content with rainfall intensity being the main factor (Allocca et al., 2015; Crosbie et al., 2012; Nasta et al., 2018; Taylor et al., 2013).

**17) Comment from Referee:** Table 3 – rainfall at bottom line is cumulative not mean

**Author's response**: Correct. We modified accordingly.

**18) Comment from Referee:** Figure 1. Confusing map. In physical (topographic) maps green is for low lands and brown for high land. Switch the color scale to fit to the customary color scale.

**Author's response:** We switched the colors according to the reviewer's suggestion.

**19) Comment from Referee:** Figure 3 enlarge text

**Author's response:** We increased the size of the text.

**20) Comment from Referee:** Figure 7 enlarge text. m-1 shouldn't be used for per month (its per meter in the SI system).

**Author's response:** We changed to "monthly recharge (mm)" to avoid misunderstanding

**21) Comment from Referee:** Figure all graphics and writing are too small. Panel C is missing.

**Author's response:** We adjusted all the graphics increasing the font size.

[Figure]

[Figure]

**Figure 1.** Schematic diagram for Mountain block and Mountain Front Recharge (Figure 2 from Wilson and Guan, 2004). The red circle represents the location of the catchment in this study.

[Figure]

**Authors' response to Reviewer 2**

**General comment**

The manuscript describes a modeling study of the spatial and temporal variation of recharge in a 2.16 km2 upland catchment in a semi-arid region. Recharge in semi-arid regions constitutes a small fraction of precipitation and is subject to a large temporal and spatial variability. Studies of this hydrological component under semi-arid conditions are relatively few although the references provided by the authors are all more than 10 years old and should thus be updated when revising the manuscript. Nevertheless, I believe that the presented study expands research on recharge in semi-arid regions and that the manuscript deserves publication after revision.

**Author's response:** We thank Reviewer 2 for the thorough review of the paper and for highlighting the lack of papers using integrated hydrologic numerical models in semi-arid environments. We responded to all the comments and revised the text to improve clarity.

**Major comments**

**1) Comment from Referee**: My major concern of the presented work relates to the calibration of the MIKE SHE model, which is inadequately carried out and described. Calibration of a hydrological model should preferably be carried out using an autocalibration method (e.g. PEST) in order to (1) identify the sensitive parameters, (2) calibrate the parameters selected for calibration using an objective method, (3) identify non-uniqueness issues and correlation among the parameters, and (4) identify uncertainty intervals of the calibrated parameter values. The process can be carried out in a more or less sophisticated procedure but in any case it makes the process transparent. The authors do not describe which parameters have been subject to calibration and it is not discussed if the resulting parameters values are reasonable based on prior knowledge of the characteristics of the site. I will encourage the authors to carry out a sensitivity and calibration analysis using an autocalibration method.

**Author's response:**

The parameters involved in the calibration process were surface roughness, detention storage, imperviousness, rooting depth, Leaf Area Index, crop coefficient, unsaturated hydraulic conductivity and water content parameters of alluvium and weathered bedrock. Although autocalibration would provide more objectivity, we consider our calibration approach to have been rigorous. We tested a wide range of parameter values supported by a large set of field data, against an objective function comprised of groundwater level and stream flow measurements, following a manual trial-and-error history matching approach.

The calibration process proceeded in an iterative manner. After each calibration run, the primary calibration parameters were examined with a variety of metrics including:

**Streamflow Calibration Metrics**

- Simulated vs Observed Average Annual flow
  - Mean Error
- Simulated vs Observed Average Monthly and Daily Flow:
  - Mean Error
  - Root Mean Squared Error
  - Correlation
  - Nash Sutcliffe Efficiency
- Graphical Plots of Simulated Streamflow Versus Observed Streamflow and Precipitation
  - Provided a qualitative measure of event correlation to observed precipitation and streamflow

**Groundwater Level Calibration Metrics**

- Simulated versus observed water levels
  - Mean Error
  - Mean Absolute Error
  - Root Mean Squared Error
  - Normalized Root Mean Squared Error
- Graphical Plot of Simulated Vs Observed Water Levels (1:1 residual plot)
  - Provided a quantitative and qualitative assessment of the residual error present at observation wells throughout the domain
- Spatial Plot of Groundwater Residuals (map)
  - Provided a quantitative assessment of water level residuals plotted in the model domain
  - Spatial patterns of fit or misfit of the model were compared against other spatial data (e.g. hydraulic conductivity, boundary conditions, land uses, surface geology) to evaluate potential correlations.

Following an assessment of these calibration targets, model parameters were revised to improve the calibration metrics. During this process our choices were informed by previous knowledge of the site gained over 20 years of investigation. To determine the final value for each of the model parameters, a wide range was explored. For example, for the hydraulic conductivity, the range for the alluvium was from $2 \times 10^{-7}$ to $5 \times 10^{-4}$ m s$^{-1}$ (from 20 to 500% of the final value), whereas for the weathered bedrock the range was from $9 \times 10^{-9}$ to $3 \times 10^{-5}$ m s$^{-1}$ (from 5 to 150% of the final value). For the saturated water content, we explored a range of values for the alluvium from 0.25 to 0.4 and for the weathered bedrock from 0.1 to 0.33.

In instances where the results were not consistent with the site conceptualization, consideration was given as to whether an alternative conceptualization would explain the results predicted by the model. Testing of alternative conceptualizations through manual simulations was chosen over optimization of single conceptualization using software such as PEST given the uncertainty in how to parameterize models in these semi-arid environments. During the calibration, important structural changes were made to the model. For example, to simulate flow in the unsaturated zone, we moved from the simpler gravity flow model to the full Richards equation because the latter better reproduced the natural processes. After few runs, we added an impervious factor to a portion of the bedrock areas where massive-bedrock ridges were observed. Given these changes and the long processing time of each run (due to the thick vadose zone), it was not possible to carry out an exhaustive optimization or sensitivity analysis. However, through the calibration process we gained semi-quantitative information about the model sensitivity to each parameter.

In particular, we found that the values of unsaturated hydraulic conductivity and water content parameters of alluvium and weathered bedrock had the strongest impact on the calibration targets. These deposits represent the upper layers of our model domain and variations in their physical and hydraulic properties control the rate of infiltration, evapotranspiration, drainage and, therefore, recharge. Another factor with a moderate impact on the generation of streamflow is the detention storage. This is because a significant amount of water from precipitation, especially at the beginning of the rainy season, infiltrates without generating runoff events at the outfall (Fig. 5). This volume of water is controlled not only by the properties of unsaturated zone (Table 2) but also by the value of detention storage assigned to each land use class (Table 1). Conversely, alterations in rooting depth, LAI and crop coefficient only elicited limited changes in streamflow.  This is because significant runoff events tend to occur as brief high-intensity precipitation events with a magnitude that far exceeds the relative amount of evapotranspiration which might occur during these events. For the same reason, though, these factors had a relatively greater effect on the volume of water available for drainage and subsequent recharge.

Our confidence about the reasonableness of the final values comes from the fact they are 1) in the same range of those present in literature (Canadell et al., 1996; Scurlock et al., 2001; Chin et al., 2000), 2) similar to those used by the Surface Water Expert Panel to model surface water flow (https://www.boeing.com/principles/environment/santa-susana/technical-reports.page), 3) in the range of those measured in the  groundwater zone during on-site investigations conducted for 20 years (Cherry et al., 2009). Further confidence regarding the calibrated model and the reasonableness of the final results is derived from the validation process. The latter is based on the comparison with previous independent recharge estimates, evidence from isotopic data sets and analysis of observed fluctuations of water level hydrographs. Moreover, we were satisfied with the fact that all the key processes at the temporal and spatial scale of interest were well represented using the model.

**Author's changes in manuscript**:

We revised the description of the approach for model calibration (line 254) and model validation (line 300). We also modified the results relative to the calibration (line 343) and validation (line 378)

**2) Comment from Referee**: My second major concern relates to the conceptualization of the system being studied. The subsurface consists of densely fractured bedrock with parallel beddings and vertical joints and faults leading to preferential flow as also emphasized by the authors at several places in the manuscript. For interpreting chloride and isotope concentration measurements preferential flow appears to be important. Furthermore, the authors have developed a conceptual model for recharge, where distribution between matrix and fractures is described (l. 469-479). The flow processes in and between the two domains are mainly based on speculation and not documented by modelling. The authors need to substantiate why two domains are not considered in their modeling approach.

**Author's response:** Actually, in a previous published paper, the roles of matrix and preferential flow were examined in detail.  Analyzing the different average Cl concentration in the vadose zone and in groundwater, Manna et al. (2017) estimated that 80% of the recharge occurs as intergranular flow in the porous matrix block and 20% as fracture flow. Therefore, we think that an EPM model, such as MIKE SHE would reproduce accurately the bulk (matrix -predominantly- and fracture) flow in the unsaturated zone. In addition, the spatial resolution (20 by 20 m cells) is such that the dense interconnected network of fractures can be approximated by an EPM model. Our confidence regarding this latter point comes also from the validation of our results, using independently derived data.

The "conceptual model" section includes findings of previous studies that are incorporated and analyzed in the light of the outcome of the present paper to create indeed a conceptual model. This is why we mention the possible occurrence of preferential flow in the deeper vadose zone and describe the potential flow mechanisms, which are not explicitly simulated with MIKE SHE but analyzed in previous studies.

**Specific comments**

**1) Comment from Referee**: l. 66-75: Please update literature review with newer references

**Author's response:** We updated the literature following also the suggestions of reviewer 1. However, we want to highlight the surprisingly lack of integrated spatially distributed models for semi-arid catchments in recent years.

**Author's changes in manuscript**: line 64-76. Text added.
Numerical hydrologic models that integrate surface water and groundwater flows have been developed to simulate the spatial and temporal distribution of surface runoff, infiltration, evapotranspiration and groundwater recharge. However, the application of nearly all such simulation tools have been limited to humid regions (Wheater et al., 2007) with minimal application to semiarid regions. Scanlon et al. (2006), in their review on recharge in semiarid areas, reported only 7 papers providing a continuous spatial distribution of recharge, out of a total of 98 studies. However, these studies investigated large areas, from 1,039,647 km$^2$ (Flint and Flint, 2007) to 60 km$^2$ (Flint et al., 2001), using a relatively coarse spatial resolution (from 72,900 m$^2$ - Flint and Flint, 2007 to 900 m$^2$ - Flint et al., 2001). In the last decade, although modeling techniques have advanced to include combined surface water-groundwater simulations, recharge in semiarid areas has been represented with a GIS approach (Hernández-Marín et al., 2018) often using remote sensing data (Wang et al., 2008; Coelho et al., 2017; Crosbie et al., 2015) or neglecting the surface water component and focusing on unsaturated zone (Levy et al., 2017; Turkeltaub et al., 2015).

**2) Comment from Referee:** l. 103-104: As fracture flow is stated to be an important flow process the authors need to substantiate why this flow process is not considered in the modelling.

**Author's response:** see response to major comment 2.

**3) Comment from Referee:** l. 153-156: Is the lateral boundary condition a closed boundary? Is the lower boundary condition based on field measurements? To which extent will it impact the modeling results? Do I understand correctly that groundwater does not contribute to stream flow and that all recharge will to deeper aquifer systems? Please elaborate on the model conceptualization.

**Author's response:**

There is a fixed head boundary conditions applied to the base and along the lateral faces of the model representing the deep groundwater flow system. The shallow water table and perched systems within the alluvium and weathered bedrock are well above this deeper water table. These heads are based on observed groundwater levels at the site and simulations based on a detailed groundwater flow model. Given that the groundwater heads associated with deep aquifer system are generally observed at relatively large depths below ground surface throughout the domain, it is expected that variations in these specific values assigned would not have a significant effect on predicted recharge values. In areas where the groundwater is observed to be closer to ground surface, the alteration of these values could potentially have a more direct effect on groundwater recharge in that a groundwater table close to the surface could rise to meet the ground surface given sufficient recharge.

It is correct that groundwater contribution to streamflow is intermittent and minimal (~ 0.1 mm $y^{-1}$ for the period of 1995-2014) and only occurs after rainfall event at the farthest downstream location of the catchment where the groundwater table rises close to the ground surface.

**Author's changes in manuscript.** Line 166 – 174. Text added: A fixed head boundary applied along the lateral sides and the bottom of the model domain (490 m asl) was used to simulate the flow to and from the deeper groundwater system, not explicitly represented in the integrated model but which extends several hundred meters (Fig. 3). These fixed heads are based on observed groundwater levels at the site and simulations based on a detailed 3-D groundwater flow model system that includes the catchment and a much larger domain beyond (AquaResource and MWH, 2007). The groundwater contribution to streamflow is minimal and intermittent (~ 0.1 mm $y^{-1}$ for the period of 1995-2014) and only occurs at the farthest downstream location of the catchment where the groundwater table rises close to the ground surface.

**4) Comment from Referee:** l. 178-179: What are the thicknesses of the two groundwater zone layers?

**Author's response:** Layer 1 has a thickness variable from 24 to 185 m (average: 109 m) whereas layer has a uniform thickness of 5 m. While layer 1 may appear very thick the 'active' part from a numerical perspective begin only when the water table is reached. Flow above that occurs in the unsaturated zone that features a finer discretization.

**Author's changes in manuscript.** Line 196-198 added to the text

**5) Comment from Referee:** l. 189: Table 2 is incomplete, unsaturated zone characteristics should also be listed.

**Author's response:** The table has been completed with porosity, field capacity, residual water content and the Van Genuchten parameters (α, n) used in the model.

The model uses three separate sets of Van Genuchten parameter to represent the pressure-saturation-hydraulic conductivity relationships; 1) alluvium, 2) weathered bedrock, 3) un-weathered bedrock. The parameters used reflect our understanding that the rock matrix transmits the largest volume of recharge, while recharge through the fractures is faster. The relationships used are biased towards the matrix response. These values were further calibrated using the groundwater level responses and the stream flow. Further rock core samples indicate a high moisture content (~80%) indicating that K is often close to $K_s$ and the hydraulic conductivity-saturation curve reflects this understanding.

**Author's changes in manuscript.** New table 2

| Hydrogeologic unit | Lithology | $K_s$ (m s$^{-1}$) | Saturation ($\theta_s$) | Field capacity ($\theta_{fc}$) | Residual Water content ($\theta_r$) | Van Genuchten parameters | | |
| --- | --- | --- | --- | --- | --- | --- | --- | --- |
| | | | | | | α | n | l |
| Alluvium | | $1\times10^{-6}$ | 0.4 | 0.25 | 0.05 | 0.021 | 1.61 | 0.5 |
| Weathered bedrock | | $2\times10^{-7}$ | 0.2 | 0.11 | 0.01 | 0.033 | 1.49 | 0.5 |
| Unweathered bedrock | Shale/Siltstone | $4.1\times10^{-10}$ to $2.3\times10^{-7}$ | 0.13 | 0.1 | 0.025 | 0.01 | 1.23 | 0.5 |
| Unweathered bedrock | Sandstone | $1\times10^{-10}$ to $1\times10^{-5}$ | 0.13 | 0.09 | 0.01 | 0.01 | 2 | 0.5 |
| Unweathered bedrock | Fault zone | $1\times10^{-9}$ to $1\times10^{-6}$ | 0.13 | 0.1 | 0.025 | 0.01 | 2 | 0.5 |

**6) Comment from Referee:** l. 205-211: Could you please be a bit more clear on how the land use are estimated.

**Author's response:** Land use classes were identified and delineated based on aerial imagery and local land cover datasets (Davis et al., 1998). Descriptions of vegetation classes and species were used in conjunction with literature values for vegetation rooting depth and leaf area indices to describe local vegetation within the model.

**7) Comment from Referee:** L140 – l. 280- : The calibration procedure needs to be elaborated and revised as described above.

**Author's response:** see main comment 1.

**8) Comment from Referee:** l. 301: Generally, I would consider a mean absolute error of 4.5 m to be rather high. Perhaps you mean root mean square error?

**Author's response:** We agree that 4.5 might be seen as high error. However, we are in a recharge area, on a topographic high with hundreds of meters of head potential. In addition, given the complex structural setting (faults located in the deeper system -not modeled), the heterogeneity of the media (porosity ranging between 2 and 20% within a meter observed in rock cores, hydraulic conductivities between $1\times10^{-5}$ and $1\times10^{-10}$ m s$^{-1}$) , the horizontal and the vertical discretization of the model, we think that 4.5 m is a reasonable mean error.

**9) Comment from Referee:** l. 303-: To me it would make more sense to compare simulated and observed hydraulic heads directly?

**Author's response:**
At the transient scale, we do not expect a good matching between simulated and observed head data. This is because of the strong subsurface heterogeneity (see response to comment 8) and because the focused recharge is soon "dissipated" through the fracture system, with head measurements in open borehole blending the contributions of several hydraulically active fractures.   However, these flow dynamics in the groundwater zone are beyond the scope of this paper.  This is why to validate the ability of the model to reproduce transient conditions, we compared the spatially-average simulated recharge against the observed heads, representing the bulk response of the system to the recharge input.

**10) Comment from Referee:** l. 316- 318: Perhaps the equivalent porous medium approach is suitable for simulation of water flow but for solute transport and the interpretation of chloride and isotopes I am not sure.

**Author's response:** Agree but this is truer for the saturated zone than for the vadose zone. As explained in the response to the main comment 2, a previous study found that at the site recharge occurs mainly as intergranular matrix flow in the vadose zone. Therefore, we think that our EPM model can be corroborated by recharge studies based on the Chloride Mass Balance method and that the isotopic composition of groundwater can be interpreted under an EPM conceptual model (especially because the ET zone is made of alluvium and weathered bedrock).

**11) Comment from Referee**: l. 352: Fig. 8a and 8b.

**Author's response:** Ops! We replaced 7b with 8b.

**12) Comment from Referee:** l. 373: Check consistency with lines 216-217.

**Author's response: Thanks.** We made it consistent.

**Spatial and temporal variability of groundwater recharge in a sandstone**

**aquifer in a semi-arid region**

[revised manuscript text omitted]

*Table 3 Stable isotope composition of rainfall.*

| Date | B/886 Rain Gauge | | | RMDF Rain Gauuage | | | Average | | |
|---|---|---|---|---|---|---|---|---|---|
| | $\delta^{18}O$ | $\delta\,^2H$ | Rainfall (mm) | $\delta\,^{18}O$ | $\delta\,^2H$ | Rainfall (mm) | $\delta\,^{18}O$ | $\delta\,^2H$ | Rainfall (mm) |
| 4/10/1994 | -4 | -19 | 3 | | | | -4.0 | -19.0 | 3 |
| 25/11/1994 | -5.2 | -18 | 6 | -5.1 | -16 | 6 | -5.2 | -17.0 | 6 |
| 13/12/1994 | -5.4 | -23 | 9 | -5.4 | -25 | 9 | -5.4 | -24.0 | 9 |
| 24/12/1994 | -10.3 | -77 | 18 | -10.1 | -69 | 18 | -10.2 | -73.0 | 18 |
| 4/1/1995 | -10.3 | -75 | 94 | -9.9 | -69 | 121 | -10.1 | -72.0 | 108 |
| 11/1/1995 | -6 | -33 | 205 | -7.4 | -45 | 202 | -6.7 | -39.0 | 203 |
| 13/01/1995 | -4.4 | -19 | 20 | -4.2 | -20 | 18 | -4.3 | -19.5 | 19 |
| 16/01/1995 | -2.8 | -11 | 12 | -2.8 | -12 | 10 | -2.8 | -11.5 | 11 |
| 26/01/1995 | -12.1 | -89 | 152 | -11.7 | -85 | 150 | -11.9 | -87.2 | 151 |
| 7/3/1995 | -6.8 | -43 | 119 | -6.4 | -40 | 109 | -6.6 | -41.5 | 114 |
| 13/3/1995 | -7.5 | -44 | NA | -7.8 | -45 | NA | -7.7 | -44.5 | NA |
| 24/3/1995 | -5.8 | -22 | NA | -5.5 | -19 | NA | -5.7 | -20.5 | NA |
| 18/5/1995 | | | | -6.4 | -42 | 34 | -6.4 | -42.0 | 34 |
| 22/6/1995 | -8.6 | -62 | 14 | -8.6 | -57 | 14 | -8.6 | -59.5 | 14 |
| Volume weighted mean and total rainfall | -8.2 | -54.2 | 650 | -8.2 | -56.2 | 691 | -8.3 | -55.2 | 689 |

Formatte

[Figure]

[Figure]

*Figure 1 Topographic map of the study area and location of the wells used for calibration (blue), water isotopes sampling*
*(red). In black the two cells where unsaturated zone water budgets were analyzed.*

[Figure]

*Figure 2 Geologic map of the study area and location of the wells used for calibration (blue), water isotopes sampling (red). In black the two cells where unsaturated zone water budgets were analyzed.*

[Figure]

[Figure]

 *Figure 3 Description of the vertical MIKE SHE model domain*

[Figure]

*Figure 4 Land use map and location of the wells used for calibration (blue), water isotopes sampling (red). In black the two*
*cells where unsaturated zone water budgets were analyzed.*

[Figure]

*Figure 5 Monthly precipitation values and comparison between simulated (green) and observed (red) runoff flow at the*
*outfall of the catchment from January 2009 to December 2011.*

[Figure]

[Figure]

*Figure 6 Comparison between simulated and observed groundwater head data for the 17 wells.*

[Figure]

*Figure 7 Comparison between the monthly recharge time series and the depth to groundwater at five locations across the*
*catchment.*

[Figure]

[Figure]

*Figure 8̶7. Distribution of average annual infiltration (a), evapotranspiration (b) and recharge (c). Dashed polygons*
*represent areas with alluvium at the surface.*

[Figure]

*Figure 9 8 Unsaturated zone water budget for ET zone from January 2004 to December 2007 for two cells representative of*
*the domain: (a) UZ-1 area with outcropping bedrock without vegetation; (b) UZ-2 area with alluvium deposit covered by*
*vegetation.*

[Figure]

*Figure 79 Comparison between the monthly recharge time series and the depth to groundwater at five locations across the*
*catchment.*

[Figure]

*Figure 10 Water isotopes plot for rainfall samples collected at two rain gauge stations and groundwater samples from 16*
*wells of the catchment.*

[Figure]

Figure 11 Conceptual model for recharge at the site. (a) Spatial 3-D conceptual model of the catchment showing where high
recharge occurs. 2-D schematic of the unsaturated zone hydrologic process during (b) dry season and (c) wet season. During
the dry season water content is between the field capacity (FC) and the permanent wilting point (PWP) and therefore is
consumed by evapotranspiration. Conversely, during the wet season, water content is above the FC and seeps into the
underlying bedrock. Numbers describe mechanisms of flow in the vadose zone: 1 is fracture flow; 2 is water flowing from
matrix into fractures; 3 is water flux from fractures into matrix; 4 is intergranular matrix flow.